# MIC26 and MIC27 cooperate to regulate cardiolipin levels and the landscape of OXPHOS complexes

Ruchika Anand[1], Arun Kumar Kondadi[1], Jana Meisterknecht[2,3,4], Mathias Golombek[1], Oliver Nortmann[1], Julia Riedel[1], Leon Peifer-Weiß[1], Nahal Brocke-Ahmadinejad[1], David Schlütermann[5], Björn Stork[5], Thomas O Eichmann[6,7], Ilka Wittig[2,3,4], Andreas S Reichert[1]

Homologous apolipoproteins of MICOS complex, MIC26 and MIC27, show an antagonistic regulation of their protein levels, making it difficult to deduce their individual functions using a single gene deletion. We obtained single and double knockout (DKO) human cells of *MIC26* and *MIC27* and found that DKO show more concentric onion-like cristae with loss of CJs than any single deletion indicating overlapping roles in formation of CJs. Using a combination of complexome profiling, STED nanoscopy, and blue-native gel electrophoresis, we found that MIC26 and MIC27 are dispensable for the stability and integration of the remaining MICOS subunits into the complex suggesting that they assemble late into the MICOS complex. MIC26 and MIC27 are cooperatively required for the integrity of respiratory chain (super) complexes (RCs/SC) and the $F_1F_o$–ATP synthase complex and integration of $F_1$ subunits into the monomeric $F_1F_o$–ATP synthase. While cardiolipin was reduced in DKO cells, overexpression of cardiolipin synthase in DKO restores the stability of RCs/SC. Overall, we propose that MIC26 and MIC27 are cooperatively required for global integrity and stability of multimeric OXPHOS complexes by modulating cardiolipin levels.

## Introduction

Mitochondria are vital cellular organelles that perform several important functions involving energy conversion, cellular metabolism, reactive oxygen species (ROS) production, heme synthesis, calcium homeostasis, and apoptosis. Mitochondrial shape is highly variable and changes constantly depending on energy demands and mitochondrial functions. Mitochondria are enclosed by a double membrane where the inner membrane (IM) folds inward to form the cristae membrane. Cristae host respiratory chain complexes and therefore are the major sites of energy conversion (Vogel et al, 2006; Wurm & Jakobs, 2006). Cristae are compositionally and functionally distinct from the rest of the IM, called inner boundary membrane (IBM) (Vogel et al, 2006; Wurm & Jakobs, 2006), presumably due to the presence of crista junctions (CJs) which are small, pore-, or slit-like openings present at the neck of a crista (Perkins et al, 1997; Mannella et al, 2001). CJs with diameter of 12–40 nm are proposed to act as diffusion barrier for ions and metabolites and therefore divide mitochondria to various sub-compartments which help streamline mitochondrial functions (Mannella, 2008; Zick et al, 2009; Mannella et al, 2013). For example, cytochrome *c* is normally trapped in the intracristal space and is released into cytosol during apoptosis after widening of CJs (Scorrano et al, 2002; Frezza et al, 2006). Recently, it was shown that CJs provide electric insulation between cristae that can display different membrane potential (Wolf et al, 2019). Cristae shape varies considerably depending on the bioenergetic demands during physiological changes and stress, including hypoxia, nutrient starvation, ROS, or induction of apoptosis (Mannella, 2006; Gomes et al, 2011; Cogliati et al, 2016; Pernas & Scorrano, 2016; Plecita-Hlavata et al, 2016; Baker et al, 2019; Dlaskova et al, 2019). The shape of cristae was suggested to govern the assembly and the stability of the respiratory chain complexes (RCs) and supercomplexes (SCs) (Cogliati et al, 2013). Aberrant cristae are present in a variety of human diseases but whether cristae ultrastructural manifestations are a cause or consequences of the pathology is often unclear. Using live-cell stimulated emission depletion (STED) super-resolution nanoscopy, we recently showed that CJs and cristae undergo dynamic remodelling in a balanced and reversible manner that is MICOS complex-dependent (Kondadi et al, 2020).

The molecular mechanisms for shaping cristae are beginning to be understood, yet an interplay of three major protein complexes, namely, OPA1, $F_1F_o$–ATP synthase, and the MICOS complex, is known to be required for formation and maintenance of cristae and CJs in

---

[1]Institute of Biochemistry and Molecular Biology I, Heinrich Heine University Düsseldorf, Medical Faculty, Düsseldorf, Germany   [2]Functional Proteomics, Sonderforschungsbereich (SFB) 815 Core Unit, Faculty of Medicine, Goethe-University, Frankfurt am Main, Germany   [3]Cluster of Excellence "Macromolecular Complexes", Goethe University, Frankfurt am Main, Germany   [4]German Center of Cardiovascular Research (DZHK), Partner Site RheinMain, Frankfurt, Germany   [5]Institute of Molecular Medicine I, Heinrich Heine University Düsseldorf, Medical Faculty, Düsseldorf, Germany   [6]Center for Explorative Lipidomics, BioTechMed-Graz, Graz, Austria   [7]Institute of Molecular Biosciences, University of Graz, Graz, Austria

Correspondence: anand@hhu.de; reichert@hhu.de

---

eukaryotic cells (Kondadi et al, 2019). OPA1 is a large dynamin-like GTPase present in the IM and has dual functions in managing mitochondrial fusion and cristae morphology (Cipolat et al, 2004). Loss of OPA1 causes severe fragmentation of mitochondria combined with reduced number of cristae that are swollen (Duvezin-Caubet et al, 2006; Song et al, 2007; Anand et al, 2013; MacVicar & Langer, 2016; Lee et al, 2017). The $F_1F_o$–ATP synthase complex well known for its classical role in converting ATP from ADP and $P_i$ using the electrochemical gradient energy across the IM also plays an important role in cristae formation (Paumard et al, 2002). The loss of the dimeric-specific subunits of $F_1F_o$–ATP synthase (Su e or Su g) leads to aberrant cristae structure with loss of the cristae rims and an arrangement of cristae as onion slices (Paumard et al, 2002). Long ribbon-like rows of $F_1F_o$–ATP synthase dimers are present at the cristae rims (Davies et al, 2011; Blum et al, 2019). An important breakthrough in understanding the mechanisms of cristae and CJs formation comes from the identification of several subunits of the MICOS ("mitochondrial contact site and cristae organizing system") complex (Rabl et al, 2009; Harner et al, 2011; Hoppins et al, 2011; von der Malsburg et al, 2011). MICOS is a large oligomeric complex required for the formation of contact sites between outer membrane and IM and formation of CJs. MICOS is highly conserved with seven bona fide subunits identified till now in mammalian system: MIC10/Minos1, MIC13/Qil1, MIC19/CHCHD3, MIC25/CHCHD6, MIC26/APOO, MIC27/APOOL, and MIC60/Mitofilin (Rampelt et al, 2017b; Quintana-Cabrera et al, 2018a). MIC10 and MIC60 are the core components of the MICOS complex because deletion of either of them causes virtually complete loss of CJs and cristae featuring as onion slices (Harner et al, 2011; Hoppins et al, 2011; von der Malsburg et al, 2011; Callegari et al, 2019; Kondadi et al, 2020). Mutations in bona fide subunits of MICOS, MIC60, MIC13, and MIC26 are found in human diseases such as Parkinson's (Tsai et al, 2018), mitochondrial encephalopathy with liver dysfunction (Guarani et al, 2016; Zeharia et al, 2016) and mitochondrial myopathy with lactic acidosis, cognitive impairment, and autistic features (Beninca et al, 2020), respectively.

Most of the research about cristae-shaping molecules, particularly the MICOS complex, are performed in baker's yeast. There is a lack of comprehensive studies addressing the individual role of mammalian MICOS subunits in managing cristae structure and mitochondrial function. We have earlier identified novel MICOS subunits in mammals, namely, MIC13, MIC26 and MIC27 using complexome profiling (Weber et al, 2013; Koob et al, 2015; Anand et al, 2016). In this study, we focus on determining the molecular role of the homologous subunits of MICOS, MIC26 and MIC27, in regulating cristae structure. MIC26/Apolipoprotein O and MIC27/Apolipoprotein O-like belong to family of apolipoproteins. Normally, apolipoproteins bind to lipids and transport them within the lymphatic and circulatory system. MIC26 was identified at elevated levels in the heart transcriptome of a diabetic model in dogs (Lamant et al, 2006) and its glycosylated (secreted) form is present at higher amounts in blood plasma of human patients of acute coronary syndrome (ACS) (Yu et al, 2012), indicating its significance to human health. We found that next to this secreted glycosylated form (55 kD), a non-glycosylated form of MIC26/APOO (22 kD) resides in the IM representing a bona fide subunit of MICOS complex (Koob et al, 2015). Only a mitochondrial form of MIC27 is observed so

far. MIC26 and MIC27 are part of the subcomplex of MICOS, MIC13-MIC10-MIC26-MIC27 where MIC27 was shown to bind to cardiolipin (CL) (Weber et al, 2013; Friedman et al, 2015), and levels of MIC26 and MIC27 are positively correlated with tafazzin, an enzyme required for cardiolipin remodelling in mitochondria (Koob et al, 2015). Both Mic26 and Mic27 are considered noncore components of the MICOS complex in yeast because their individual deletion does not lead to drastic cristae alterations (Harner et al, 2011; Hoppins et al, 2011; von der Malsburg et al, 2011; Zerbes et al, 2016; Eydt et al, 2017). In yeast, Mic26-Mic27 antagonism and cardiolipin are required for assembly of Mic10 oligomers (Rampelt et al, 2018). Partial knockdown of MIC26 and MIC27 in mammalian cells also causes only moderate cristae defects (Weber et al, 2013; Koob et al, 2015). Intriguingly, Western blots revealed that steady-state levels of MIC26 and MIC27 are reciprocally regulated as depletion or overexpression of one of them is always accompanied with increase or decrease in protein level of the second protein, respectively, showing an antagonistic regulation (Koob et al, 2015; Rampelt et al, 2018). This antagonistic regulation makes it difficult to infer their individual role using only single deletion mutants. This could mean that the moderate defects observed in cristae structure upon single depletion of MIC26 or MIC27 might occur because of their partial overlapping function and compensation by the second protein. Therefore, to determine the individual as well as overlapping functions of MIC26 and MIC27, we decided to knockout MIC26 and MIC27 individually and in combination in mammalian cell lines. We found that double knockout (DKO) cell lines lacking MIC26 and MIC27 show accumulation of aberrant cristae and reduced $F_1F_o$–ATP synthase activity and cellular respiration. This was accompanied by decreased steady-state levels of OXPHOS complexes with reduced cardiolipin levels, and complexome profiling showed a partial dissociation of $F_1$ subunits from the $F_1F_o$ monomer in the DKO of MIC26 and MIC27. Overall, we suggest that MIC26 and MIC27 act in cooperation to regulate cristae structure and the global integrity of respiratory chain complexes and supercomplexes (RCs and SCs) and $F_1F_o$–ATP synthase complex.

# Results

### MIC26 and MIC27 are reciprocally regulated at the posttranscriptional level

To determine the molecular role of MIC26 and MIC27 in maintaining cristae structure, we obtained single as well as double knockout (DKO) cells for MIC26 and MIC27. These cells were generated in haploid cell lines, HAP1 cells. Single knockouts (SKOs) were prepared using non-homologous end joining CRISPR-Cas method, yielding a 1-bp insertion and an 8-bp deletion in exon 3 of MIC26 and MIC27, respectively. These insertions or deletions yielded frame shift and missense mutations causing premature termination of transcription and complete loss of proteins determined by Western blot analysis using antibodies against endogenous MIC26 or MIC27 (Fig 1A). The DKO cells lacking MIC26 and MIC27 were obtained by targeting MIC27 using CRISPR-Cas that causes a 160-bp deletion in exon 3 of MIC27 in MIC26 KO cells. DKO cells lack the full-length

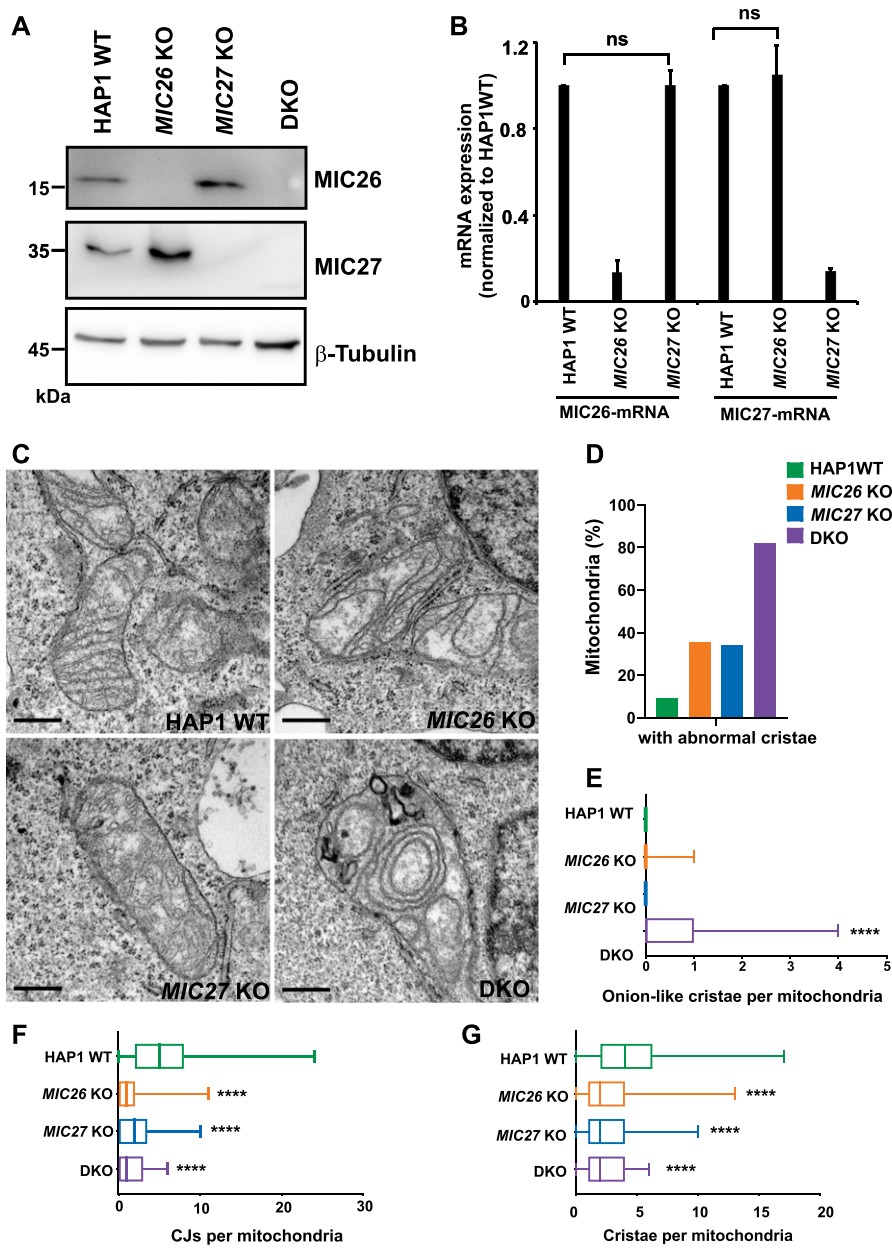

**Figure 1. MIC26 and MIC27 cooperatively determine cristae morphology and are required for formation of crista junctions (CJs).**

**(A)** Western blots from total cell lysates from HAP1 WT, *MIC26* KO, *MIC27* KO or double knockout (DKO) cells show loss of respective protein and increased level of the respective other protein (reciprocal regulation). DKO cells lacking *MIC26* and *MIC27* show virtually a complete loss of both full-length proteins. **(B)** Quantification from qRT-PCR using HAP1 WT, *MIC26* KO or *MIC27* KO cells and probed for the mRNA levels of MIC26 or MIC27 using specific primers. The house-keeping genes HPRT1 and GAPDH were used as controls. Data from three independent experiments represented as mean ± SEM. *P*-values calculated using *t* test show no significant differences (ns). **(C)** Representative images from electron microscopy in HAP1 WT, *MIC26* KO, *MIC27* KO, or DKO cells show accumulation of concentric cristae in DKO cells and loss of CJs in single knockouts (SKOs) and DKO cells. Scale bar 0.5 $\mu$m. **(D)** Bar graph show the percentage of mitochondria that have abnormal cristae in respective cell lines. Data from total of 60–90 mitochondrial sections of two independent experiments are represented. **(E)** Box plot showing the number of concentric onion-like cristae per mitochondrial section. DKO cells show high accumulation of concentric cristae. Data from total of 60–90 mitochondrial section of two independent experiments. ****$P$-value ≤ 0.0001 indicated in the plot shows comparison between WT and DKO. *t* test was used for statistical analysis. **(F)** Box plot showing the number of CJs per mitochondrial section in respective cell lines. SKOs and DKO cells have significant reduction in CJs per mitochondrial section. Data from total of 60–90 mitochondrial section of two independent experiments. ****$P$-value ≤ 0.0001. *P*-value indicated in the graph show comparison between WT and respective cell lines. Comparisons between *MIC26* KO and *MIC27* KO as well as DKO with *MIC26* KO were not significantly different (*P*-value > 0.5). Comparison between *MIC27* KO and DKO show slightly significant difference (*P*-value = 0.04). *t* test was used for statistical analysis. **(G)** Box plot showing the number of cristae per mitochondrial section in respective cell lines. SKOs and DKO cells have significant reduction in cristae per mitochondrial section. Data from total of 60–90 mitochondrial sections of two independent experiments. ****$P$-value ≤ 0.0001. *P*-value indicated in the graph show comparison between WT and respective cell lines. Comparisons between *MIC26* KO and *MIC27* KO as well as DKO with *MIC26* KO or *MIC27* KO were not significantly different (*P*-value > 0.5). *t* test was used for statistical analysis.

MIC27 protein, although occasionally we observe a very faint band at lower molecular weight, which likely occurs because of alternative splicing and skipping the deleted exon. The steady-state levels of MIC26 and MIC27 were increased in *MIC27* and *MIC26* KO cells, respectively, corroborating the reciprocal and antagonist regulation reported earlier (Koob et al, 2015) (Figs 1A and 3C). To check whether this regulation at steady-state levels is determined by a transcriptional regulation, we performed quantitative real time PCR (qRT-PCR) using primers specific to mRNA of MIC26 and MIC27 and compared them with the house-keeping genes, HPRT1 and GAPDH in *MIC26* and *MIC27* SKOs. We found that the mRNA levels of MIC26 and MIC27 are not significantly altered in *MIC27* and *MIC26* KO cell lines, respectively (Fig 1B), implying that this change in the

protein levels of MIC26 or MIC27 occurs at a posttranscriptional level.

## MIC26 and MIC27 are cooperatively required to maintain mitochondrial cristae ultrastructure

Depletion of MIC26 or MIC27 was reported to show moderate defects in cristae structure accounting for these subunits to be considered as noncore subunits of the MICOS complex (Weber et al, 2013; Koob et al, 2015). However, as stated above, steady-state levels of MIC26 or MIC27 are reciprocally maintained, and deletion of one of these subunits is always accompanied by concomitant increase in the respective other protein (Fig 1A). Therefore, it is possible that cristae

defects observed because of the deletion of single subunit is masked or caused by simultaneous up-regulation of the other homologous subunit. This could also point to a partial or complete redundant function of MIC26 and MIC27 in regulating cristae morphology. Hence, to determine the combined molecular role of MIC26 and MIC27 in regulating cristae architecture, we analyzed the ultrastructure of mitochondria using electron microscopy in SKOs and DKO cells of *MIC26* and *MIC27*. Control HAP1 cells show typical lamellar cristae that are arranged parallel to each other and connected to the IBM via CJs (Fig 1C). At a first glance, only cristae from DKO cells show the presence of characteristic MICOS-specific cristae defects where cristae are arranged as onion stacks in the mitochondria (Fig 1C). We determine the percentage of mitochondria that contain abnormal cristae in all the cell lines and found that while SKOs show accumulation of aberrant cristae, much higher percentage of mitochondria show abnormal cristae in DKO cells (Fig 1D). In addition, the appearance of aberrant onion-like cristae structures was significantly increased only in DKO cells compared with control or SKOs (Fig 1E). Longitudinal and vesicular

cristae were more prevalent in *MIC26* KO and *MIC27* KO, respectively (Fig 1C). We performed a detailed analysis for various parameters using the electron micrographs from all cell lines. We observed that both SKOs and DKO cells show significantly reduced CJs per mitochondrial section compared with the controls (Fig 1F). In addition, the number of cristae per mitochondrial section was significantly reduced in all the KO cells compared with control, with a slightly more pronounced reduction in DKO cells compared with SKOs (Fig 1G). Since MIC26 and MIC27 only partially complement each other regarding cristae defects in their respective SKOs and as the most pronounced effects are seen in DKO cells (Fig 1C–G), we can conclude that both MIC26 and MIC27 are functionally overlapping, yet not fully redundant.

### Simultaneous deletion of MIC26 and MIC27 causes reduced respiration and mitochondrial fragmentation

Next, we asked how cristae defects associated with SKOs and DKO cells lacking *MIC26* and/or *MIC27* affect mitochondrial function. First, we analyzed oxygen consumption rates to determine the

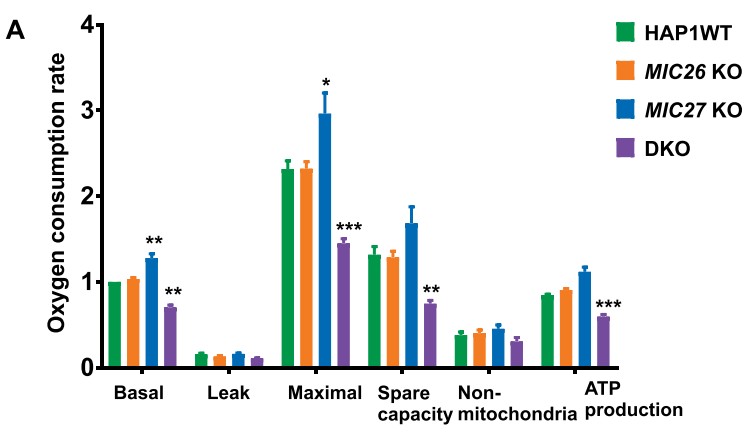

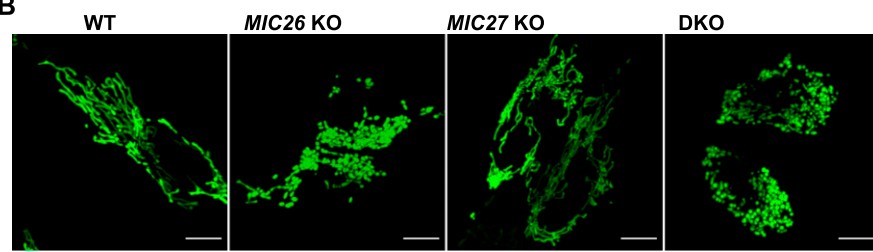

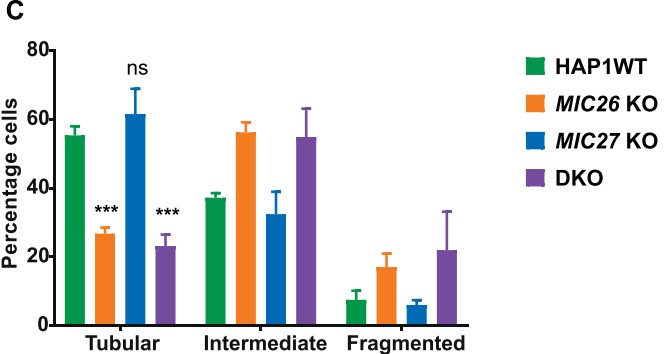

**Figure 2. Mitochondrial respiration is impaired and mitochondria show fragmentation in double knockout (DKO) cells lacking *MIC26* and *MIC27*.**
**(A)** Oxygen consumption rates (pmol $O_2$/s, normalized for cell numbers by Hoechst staining), including basal respiration (Basal), proton leak, maximal respiration (Maximal) after uncoupling by FCCP, spare respiratory capacity (Spare capacity), non-mitochondrial respiration (Non-mitochondrial), and ATP production is shown for HAP1 WT, *MIC26* KO, *MIC27* KO, or DKO cells. Data are normalized to basal respiration from HAP1 WT and the mean ± SEM from four independent experiments is shown. DKO cells lacking *MIC26* and *MIC27* show reduced respiration, whereas *MIC27* KO show slight but significant increase compared with HAP1 WT. *$P$-value ≤ 0.05, **$P$-value ≤ 0.01, ***$P$-value ≤ 0.001 ($t$ test). For comparison of basal respiration, one sample $t$ test was performed. **(B)** Representative confocal images of mitochondria from HAP1 WT, *MIC26* KO, *MIC27* KO, or DKO cells show mitochondrial fragmentation in *MIC26* KO and DKO cells. **(C)** Quantification of percentage of cells having tubular, intermediate, or fragmented mitochondrial morphology in HAP1 WT, *MIC26* KO, *MIC27* KO, or DKO cells. Data show mean ± SEM from three independent experiments. $t$ test was used for comparison of percentage of cells having tubular mitochondria in *MIC26* KO, *MIC27* KO, or DKO cells with HAP1 WT. ***$P$-value ≤ 0.001.

cellular respiration in SKOs and DKO cells. We found that basal and maximal mitochondrial respiration in DKO cells are significantly reduced compared with control cells (Fig 2A). Although respiration in *MIC26* KO cells was not grossly altered despite the ultrastructural defects, we observed a slight but significant increase in basal and maximal respiration of *MIC27* KO compared with controls (Fig 2A). This indicates that in these cell lines, only the simultaneous deletion of MIC26 and MIC27 can sufficiently affect the mitochondrial respiration, whereas the respiration in SKOs can be compensated (or even enhanced) due to concomitant overexpression of the second protein. To check for the specificity of respiration defects in DKO cells, we stably reintroduced MIC26 and/or MIC27 in these cell lines (Fig S1A) and found that overexpression of both MIC26 and MIC27 could significantly restore the oxygen consumption rates of DKO (Fig S1B). Mitochondrial morphology is another determinant of cellular or mitochondrial dysfunction (Duvezin-Caubet et al, 2006). Thus, we checked the mitochondrial morphology in SKOs and DKO cells of *MIC26* and *MIC27* and found that *MIC26* KO and DKO cells show a similar increase in the extent of mitochondrial fragmentation (Fig 2B and C), whereas *MIC27* KO cells show normal mitochondria which were comparable to control cells (Fig 2B and C). This indicates a role of MIC26 in regulating mitochondrial morphology that cannot be compensated by MIC27 and is therefore independent of MIC27 function.

## MIC26 and MIC27 are dispensable for the incorporation of other MICOS subunits into MICOS complex

To determine the molecular basis for the reduction in the amount of CJs that we observed in SKOs and DKO cells of *MIC26* and *MIC27* (Fig 1F), we asked whether MIC26 and MIC27 influence the MICOS complex and/or the assembly of the $F_1F_o$–ATP synthase, two important players determining cristae morphogenesis in baker's yeast (Rabl et al, 2009; Eydt et al, 2017; Rampelt et al, 2017a). We characterized the immunostaining pattern of MIC26 and MIC27 using STED super-resolution nanoscopy. It was shown that MIC10 or MIC60, the core MICOS subunits, show an equally spaced rail-like arrangement of punctae across the mitochondrial length (Jans et al, 2013; Stoldt et al, 2019; Kondadi et al, 2020). The staining pattern of MIC26 and MIC27 resembles the MICOS-specific punctae and appeared similar to that of MIC60 and MIC10 indicating that MIC26 and MIC27 assembled in a regular rail-like fashion characteristic of the MICOS complex (Figs 3A and 4A). Next, we asked how the loss of MIC26 and/or MIC27 affects the stability or integrity of the MICOS complex. First, we determined the steady-state levels of other MICOS subunits in SKOs and DKO cells. We did not observe any significant and consistent change in steady-state levels of other MICOS subunits in SKOs or DKO cells apart from increase in levels of MIC26 and MIC27 in *MIC27* and *MIC26* KO cells (reciprocal regulation) (Fig 3B and C). Only a minor increase in the amount of MIC25 was seen in DKO cells (Fig 3C). We conclude that MIC26 and MIC27 are not required for the stability of other MICOS subunits. Second, we checked how the loss of MIC26 and/or MIC27 affects the localization or arrangement of core components of MICOS complex, namely, MIC60 or MIC10 in the mitochondria using STED super-resolution nanoscopy. Consistent with earlier reports (Jans et al, 2013; Stoldt et al, 2019; Kondadi et al, 2020), MIC60 and MIC10 showed a MICOS-

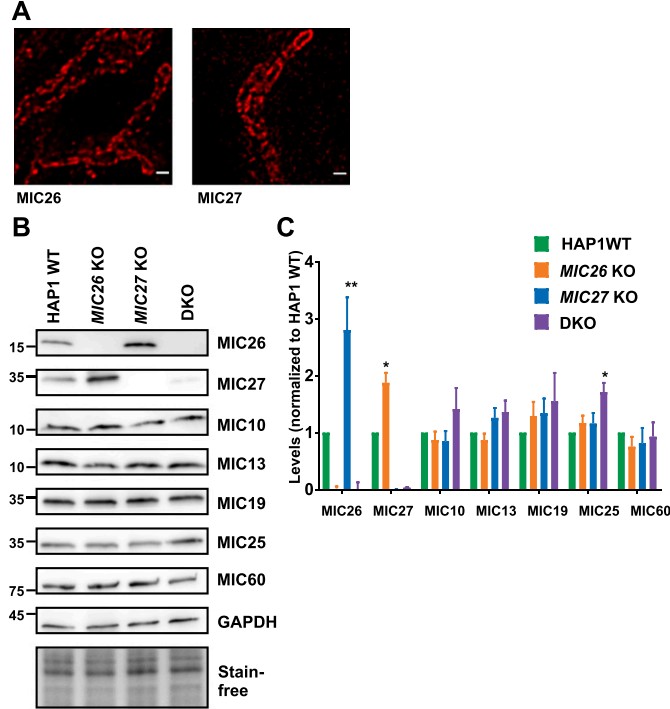

**Figure 3. MIC26 and MIC27 are not required for the stability of other MICOS subunits.**
**(A)** Representative STED super-resolution images of MIC26 or MIC27 in control cells show the punctae rail-like arrangement within mitochondria that resemble the staining from MIC60 or MIC10 (see also Fig 4A). Scale bar 0.5 $\mu$m. **(B)** Western blots of total cell lysates from HAP1 WT, *MIC26* KO, *MIC27* KO, or double knockout (DKO) cells probed for various subunits of the MICOS complex. **(C)** Densitometric quantification of Western blots from four independent experiments (mean ± SEM) in HAP1 WT, *MIC26* KO, *MIC27* KO, or DKO cells that are normalized to levels of each MICOS subunits to the HAP1 WT. Except MIC26 or MIC27 levels (showing reciprocal change), steady-state levels of other MICOS subunits were not drastically reduced in single knockouts or DKO cells of *MIC26* and *MIC27*. One sample *t* test was used for comparison. Marginal increase in MIC25 was found in DKO cells. *$P$-value ≤ 0.05. **$P$-value ≤ 0.01.

specific pattern of regularly arranged punctae across the mitochondrial length in WT HAP1 cells (Fig 4A). This staining pattern was indistinguishable in SKOs or DKO cells lacking *MIC26* and/or *MIC27* indicating that MIC26 and MIC27 are dispensable for the formation of the MICOS scaffold in the IBM of mitochondria (Fig 4A). Third, to determine the composition or the integrity of the whole MICOS in our KO cell lines, we performed complexome profiling. Upon carefully comparing the complexome profiles of mitochondria isolated from SKOs and DKO cells with the controls, we found that MIC26 or MIC27 can assemble into high molecular weight complexes independent of each other, and none of them is required for incorporation of other MICOS subunits into these higher molecular weight complexes (Figs 4B and S2). In DKO mitochondria, the remaining MICOS subunits were shifted to a lower molecular weight complex than control cells (Figs 4B and S2), which is also seen consistently in blue-native (BN)–PAGE (Fig 4C). The observation that the amount of the high molecular weight MICOS complex, but not that of the lower molecular weight MICOS complex, is slightly reduced in *MIC26* KO and DKO cells can be attributed to the loss of the respective subunits (Figs 4B and S2). However, we cannot exclude

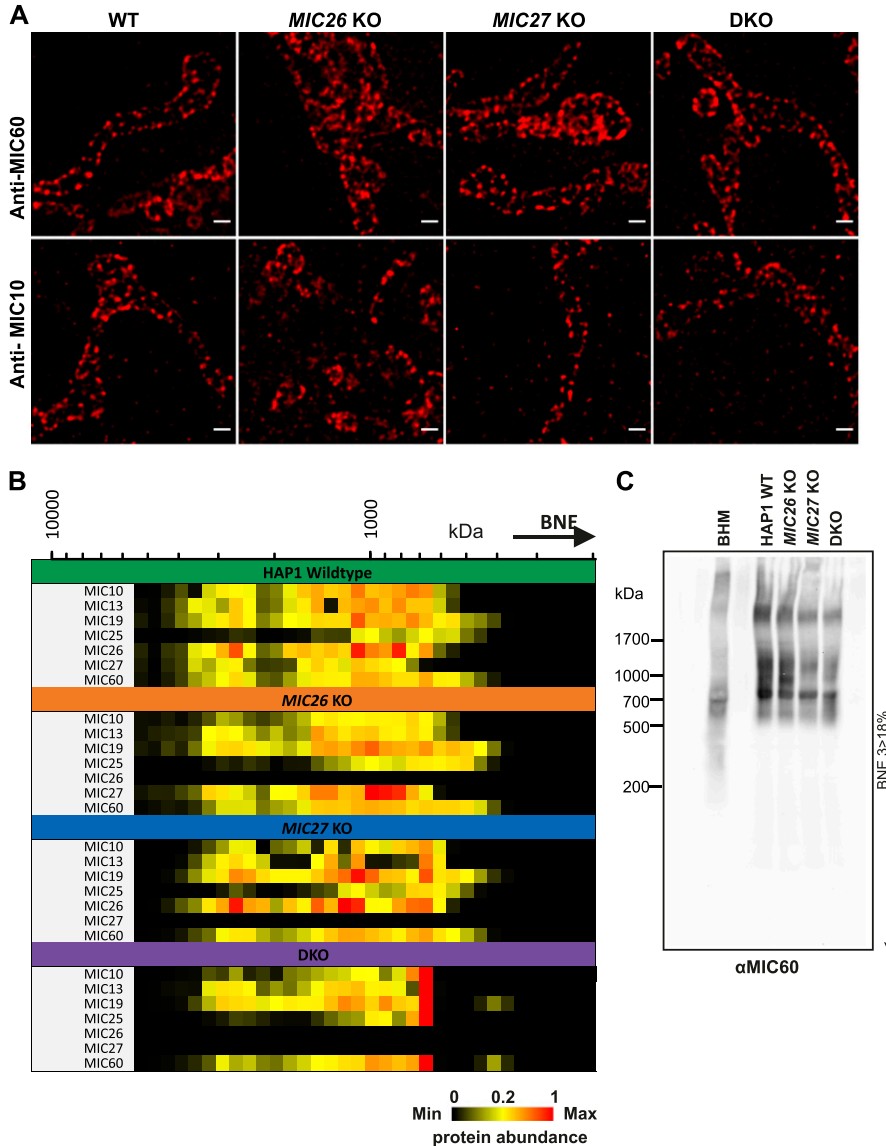

**Figure 4. MIC26 and MIC27 are dispensable for the spatial arrangement of MIC10 and MIC60 in mitochondria and the incorporation of other MICOS subunits into the complex.**
**(A)** Representative images of endogenous staining of MIC60 or MIC10 in HAP1 WT, *MIC26* KO, *MIC27* KO, or double knockout (DKO) cells using STED super-resolution nanoscopy show that the rail-like punctae arrangement of MIC60 or MIC10 remain unaltered in single knockouts or DKO cells. **(B)** Complexome profiling data representing the heat map of abundance of occurrence of MICOS subunits in isolated mitochondria from HAP1 WT, *MIC26* KO, *MIC27* KO, or DKO cells show that cluster of MICOS complex shifts to a lower molecular weight in DKO mitochondria but the remaining subunits remain associated to this complex. **(C)** Blue-native gel electrophoresis blotted for anti-MIC60 show MICOS complex in single knockouts or DKO cells lacking *MIC26* and/or *MIC27*. BHM, bovine heart mitochondria.

that MIC26 and MIC27 play a role in the integrity of the high molecular weight MICOS complex. Yet, as the stability of MICOS subunits and their incorporation in MICOS is not grossly perturbed, we conclude that MIC26 and MIC27 do not play a role in bridging the two subcomplexes, MIC19-25-60 and MIC13-10-26-27, of the MICOS complex as was reported for MIC13 (Guarani et al, 2015; Anand et al, 2016). This would be consistent with MIC26 and MIC27 assembling into the MICOS complex at a rather late stage and fits to the observation that the steady-state levels of all other MICOS subunits are not reduced upon loss of MIC26 and/or MIC27.

### MIC26 and MIC27 are required for the assembly and stability of the monomeric $F_1F_o$–ATP synthase

The oligomerization of the $F_1F_o$–ATP synthase is associated with proper formation of cristae, in particular by inducing a positive curvature at the rim of a crista (Strauss et al, 2008; Rabl et al, 2009). To further determine whether loss of MIC26 and MIC27 affect the oligomerization and stability of the $F_1F_o$–ATP synthase and thus contribute to the cristae defect observed earlier, we checked the activity and oligomerization of $F_1F_o$–ATP synthase in SKOs and DKO cells. For this, we performed in-gel ATP hydrolysis activity assays after native gel electrophoresis to reveal $F_1F_o$–ATP synthase oligomers, dimers, monomers, or active $F_1$ subcomplexes. In this assay, we found a striking reduction of the overall intensity of $F_1F_o$–ATP synthase activity in DKO cells compared with SKOs and control cells (Fig 5A, left panel). This was not due to unequal loading as confirmed by Western blot analysis and Ponceau S staining (Fig 5A, right panel). The relative amounts of $F_1F_o$–ATP synthase complex found as oligomers, dimers, or monomers were not significantly altered in DKO cells compared with the SKOs or control cells (Fig 5B), indicating no obvious defect in the oligomerization of monomeric

**Figure 5. MIC26 and MIC27 are cooperatively required for the stability and assembly of the F₁Fₒ–ATP synthase complex.**

**(A)** Blot showing the in-gel activity of F₁Fₒ–ATP synthase using isolated mitochondria of HAP1 WT, *MIC26* KO, *MIC27* KO, or double knockout (DKO) cells that were solubilized with increasing concentration of digitonin (g/g). The blot show oligomers, dimers, and monomers forms of F₁Fₒ–ATP synthase. The intensity (or activity) was reduced in DKO cells. The same mitochondrial lysate was blotted on SDS–PAGE to probe for equal loading among the samples. **(B)** The quantification of ratio of oligomers or dimers or monomers of F₁Fₒ–ATP synthase to the total intensity in the lane specific for 0.75 g/g digitonin was calculated from three independent experiments (mean ± SEM) show no significant difference among them in single knockouts or DKO cells lacking *MIC26* and/or *MIC27* compared with HAP1 WT. ns = *P*-value > 0.05 (nonsignificant).

$F_1F_0$–ATP synthase complexes despite the reduced overall activity (Fig 5A). This result is distinct to data from baker's yeast, where loss of Mic26 showed reduced oligomerization of the $F_1F_0$–ATP synthase (Eydt et al, 2017) pointing to a difference in regulation of $F_1F_0$–ATP synthase oligomerization in mammals as compared with fungi. The reduced staining in the in-gel activity assay could be attributed to either the loss of the functionality or reduced amounts of the $F_1F_0$–ATP synthase complex. To check this, we performed BN–PAGE and probed with an antibody specific for $F_1F_0$–ATP synthase complex, ATP5D, using mitochondrial lysates from SKOs and DKO cells lacking *MIC26* and *MIC27* and control cells (Fig 5C). In BN–PAGE, we found a considerable decrease in the amount of the monomeric $F_1F_0$–ATP synthase in the DKO as compared with SKOs and control cells (Fig 5C). This indicates that the reduced activity of $F_1$ in the $F_1F_0$–ATP synthase complex in DKO cells is due to reduced amount of the $F_1F_0$–ATP synthase complex. Moreover, we observed an accumulation of a low molecular weight band specifically in mitochondria derived from DKO cells which we attribute to a detached/non-assembled $F_1$ subcomplex. We also checked the steady-state levels of several subunits of $F_0$- or $F_1$-moieties in knockout cell lines and found no drastic or significant change in either of the subunits tested (Fig 5D), showing that the reduced amount of the $F_1F_0$–ATP synthase complex does not arise because of decrease in the supply of the subunits that we tested. To check this in more detail and test whether the assembly of the $F_1F_0$–ATP synthase complex is impaired, we analyzed our mitochondrial complexome profiling data from all the knockout cell lines. A comparison between the heat map for the $F_1F_0$–ATP synthase complexes in our control and knockout cell lines showed a drastic decrease in the amount of the $F_1F_0$–ATP synthase complex in DKO mitochondria (Figs 5E and S3), whereas *MIC27* KO mitochondria appeared to have even slightly higher levels of this complex. In WT mitochondria, the subunits of the monomeric $F_1F_0$–ATP synthase cluster at expected size of around 600 kD (Figs 5E and S3). In DKO mitochondria, apart from this complex, we find that $F_1$ subunits were accumulating at a lower molecular weight (Figs 5E and S3). This indicates an impaired assembly of the monomeric $F_1F_0$–ATP synthase complex or a partial disassembly of $F_1$ subunits upon solubilization from this complex in DKO cells consistent with the BN–PAGE data described above (Fig 5C). Overall, we conclude that in mammalian cells, MIC26 and MIC27 are dispensable for the oligomerization of the $F_1F_0$–ATP synthase complex but are rather specifically required for the integration of $F_1$ subunits into the monomeric $F_1F_0$–ATP synthase which stabilize the $F_1F_0$–ATP synthase complex.

### MIC26 and MIC27 regulate the stability of respiratory chain complexes and supercomplexes

Mitochondrial cristae shape mediated by OPA1 was suggested to regulate the assembly of respiratory chain supercomplexes (SCs) and respiratory capacity (Cogliati et al, 2013). To further decipher the

basis for the impaired respiration observed earlier (Fig 2A) and to test whether there is a link between MIC26/MIC27 and respiratory chain complexes (RCs) and SCs formation, we performed BN–PAGE from mitochondria isolated from control cells, SKOs, and DKO cells lacking *MIC26* and/or *MIC27* and probed with the antibodies specific for complexes I, III, and IV. We find that the individual complexes and their higher associations into SCs were drastically reduced in the DKO compared with the controls or the SKOs (Fig 6A, left panel). This was not due to reduced loading among the cell lines (Fig 6A, right panel). To further substantiate the role of MIC26 and MIC27 in RCs and SCs formation, we analyzed our complexome profiling data and found that there was a clear and drastic reduction in RCs and SCs in DKO cells (Figs 6B and S4), consistent with the BN–PAGE (Fig 6A). *MIC27* KO cells show slight increase in RCs and SCs that could be attributed to increased respiration in these cells (Fig 2A). Moreover, we observed certain changes that are specific to individual SKOs of *MIC26* and *MIC27*, for example, a higher molecular weight distribution of complex II in *MIC27* KO but not in other cell lines (Figs 6B and S4). We then examined the steady-state levels of marker proteins from the various RCs using Western blot analysis and found that their levels were not significantly reduced in the DKO cells (Fig S5A and B), indicating that the decrease in the levels of RCs or SCs is not due to an overall reduction in the subunits of the RCs. However, we cannot rule out that some key assembly factors causing destabilization of RCs or SCs are specifically affected in these knockout cell lines. Overall, we provide several lines of evidence that cristae defects caused by deletion of *MIC26* and *MIC27* are associated with reduced steady-state levels of fully assembled RCs and SCs. As OPA1 levels were shown to determine the assembly of SCs and respiration (Cogliati et al, 2013), we wanted to check the levels of the distinct OPA1 forms (long or short-OPA1) in the KO cells to determine whether a change in OPA1 levels could be a cause of reduced levels of SCs (or RCs). Typically, five prominent forms of OPA1 (two long forms, a/b, and three short form, c/d/e) are observed in cultured mammalian cells arising from various splice variants and two proteolytic cleavage sites (for OMA1 or YME1L) (Duvezin-Caubet et al, 2006, 2007; Anand et al, 2013; MacVicar & Langer, 2016), contributing to a complex regulation. We did not observe any consistent or drastic change in the amount or the prevalence of OPA1 forms (long or short forms) in any of our knockout cells compared with control cells (Fig S6A and B), indicating no obvious difference in OPA1 levels in cells deleted for *MIC26* or *MIC27*. From this, we conclude that the reduced levels of RCs or SCs in the knockout of *MIC26* or *MIC27* occurs independent of OPA1 regulation.

### Cardiolipin levels are reduced upon double deletion of MIC26 and MIC27

MIC26 and MIC27 are apolipoproteins that normally bind lipids and MIC27 was shown to bind to cardiolipin (CL) in vitro (Weber et al,

---

*t* test was used for statistical analysis. **(C)** Blue-native gel electrophoresis of isolated mitochondria from HAP1 WT, *MIC26* KO, *MIC27* KO, or DKO cells that were solubilized with increasing concentration of digitonin (g/g) is blotted and probed for $F_1F_0$–ATP synthase subunit, ATP5D show reduced staining in DKO cells lacking MIC26 and MIC27 with concomitant appearance of lower molecular weight complex ($F_1$). **(D)** Western blot from the lysate of HAP1 WT, *MIC26* KO, *MIC27* KO, or DKO cells were probed with antibodies specific to various subunits of $F_1F_0$–ATP synthase complex, do not show any consistent change in either of them in single knockouts or DKO cells. **(E)** Complexome profiling of isolated mitochondria from HAP1 WT, *MIC26* KO, *MIC27* KO, or DKO cells for the $F_1F_0$–ATP synthase complex showing the heat map of occurrence of subunits of $F_1F_0$–ATP synthase. $F_1F_0$–ATP synthase complex is reduced and subunits of the F1 part are partially dissociated from the complex in DKO cells lacking *MIC26* and *MIC27*.

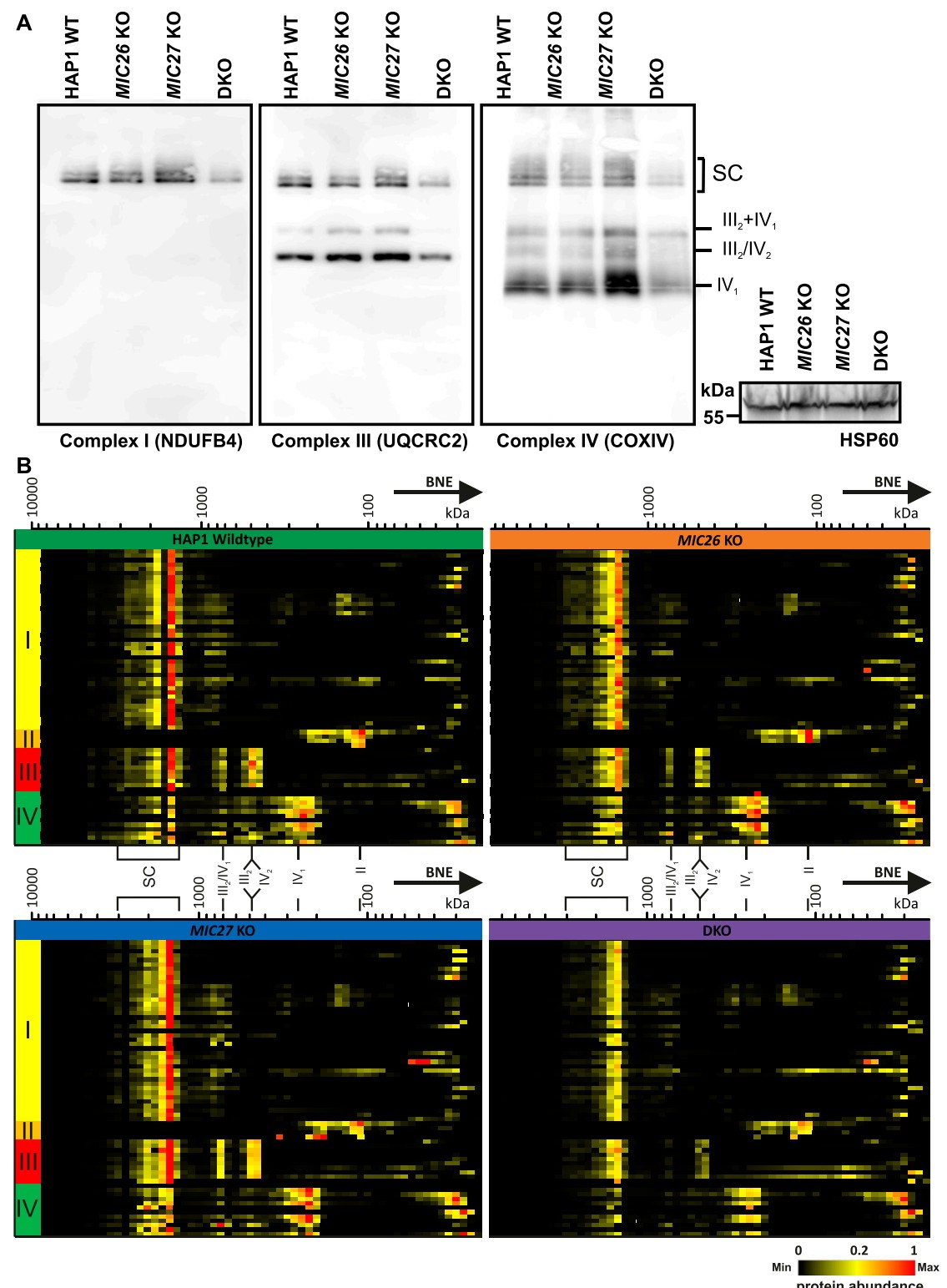

**Figure 6. MIC26 and MIC27 are required for the stability of respiratory chain (super) complexes.**

**(A)** Blue-native gel electrophoresis for isolated mitochondria from HAP1 WT, *MIC26* KO, *MIC27* KO, or double knockout (DKO) cells that were solubilized and blotted for antibodies specific for complex I (NDUFB4), complex III (UQCRC2), or complex IV (COX1V) show reduced staining of respiratory chain complexes (RCs) and their higher assemblies (supercomplexes). The same mitochondrial lysate was blotted on SDS–PAGE to probe for equal loading among the samples. **(B)** Complexome profiling of isolated mitochondria from HAP1 WT, *MIC26* KO, *MIC27* KO, or DKO cells for respiratory chain complexes (RCs)/supercomplexes (SCs) showing the heat map of occurrence of subunits of respiratory chain complexes which were reduced in DKO cells lacking *MIC26* and *MIC27*.

2013). Cardiolipin is required for the formation and stabilization of RCs and SCs (Zhang et al, 2002; Pfeiffer et al, 2003; Bottinger et al, 2012). We hypothesize that the deletion of *MIC26* and *MIC27* (or MICOS) possibly alters the cardiolipin composition of the lipid bilayer of mitochondria and thereby affects the integrity of RCs and/or SCs. Therefore, we determined the levels of cardiolipin in SKOs and DKO cells of *MIC26* and *MIC27* using mass spectrometry. We found significantly reduced levels of cardiolipin in DKO and *MIC26* KO cells, whereas they remain normal in *MIC27* KO cells (Fig 7A). This overall reduction of CL in DKO and *MIC26* KO cells was not due to specific cardiolipin species being affected predominantly (Fig 7B) but rather appeared to occur for all CL species, indicating no major defect in cardiolipin remodelling. To determine whether the reduced levels of cardiolipin affect the stability of the RCs and SCs in DKO cells, we stably overexpressed cardiolipin synthase (CRLS1) in DKO cells and analyzed RCs and SCs using BN-PAGE. As a control, we stably overexpressed MIC26 or/and MIC27 in DKO cells (Fig S1A) and found that overexpression of both MIC26 and MIC27 rescued the levels of RCs and SCs in DKO cells comparable to the control conditions (Fig 7C). Stable overexpression of CRLS1 in DKO cells could restore the stability of the RCs and SCs compared with the DKO cells with empty vector. In addition, we wanted to analyze whether overexpression of CRLS1 that restored the stability of RCs and SCs could then rescue the reduced respiration of DKO cells. Indeed, we found a significant increase in maximal respiration and ATP production upon overexpression of CRLS1 in DKO cells compared to DKO cells with empty vector (Fig S1B) showing that reduced cardiolipin in DKO cells directly affect the stability of RCs and SCs as well as respiration of DKO cells. Overall, we propose a model that the homologous subunits of MICOS, MIC26 and MIC27, are cooperatively required to modulate the levels of cardiolipin in mitochondria and influence the general stability and integrity of the respiratory chain (super) complexes and $F_1F_o$–ATP synthase complex (Fig 7D).

## Discussion

MIC26 and MIC27 are homologous proteins of the MICOS complex whose steady-state protein levels are reciprocally regulated. Here, we found that SKOs of *MIC26* and *MIC27* show moderate cristae defects when compared with DKO cells that show a clear increase in extent of cristae defects with accumulation of onion-like cristae. Therefore, MIC26 or MIC27 can partially complement each other, yet also fulfil functional roles that cannot be fully compensated by the respective other subunit. This shows that the coordinated function of MIC26 and MIC27 is required for the proper formation of CJs and indicates cooperation between MIC26 and MIC27 to regulate cristae structure. How do MIC26 and MIC27 affect MICOS complex function and thereby CJ formation? Unlike the loss of MIC60, MIC10, or MIC13 that led to destabilization of either the whole or part of the MICOS subcomplex, MIC26 and MIC27 were not required for the stability of the known remaining subunits of the MICOS complex and their incorporation into higher molecular weight complexes. STED super-resolution nanoscopy showed no change in the pattern of localization of MIC60 or MIC10 in the DKO cells lacking *MIC26* and *MIC27*,

reiterating that they do not grossly perturb the spatial organization of the MICOS complex within mitochondria and perhaps assemble later than MIC10 or MIC60 during the formation of MICOS complex. This is also consistent with our recent finding that the staining pattern of MIC60 remains unperturbed in *MIC10* KO cells that have virtually complete loss of CJs (Kondadi et al, 2020) and imply that MIC60 acts as a priming factor for the formation of CJs and remain associated at sites or hotspots of the CJ formation (Friedman et al, 2015; Stoldt et al, 2019; Kondadi et al, 2020). Mitochondrial morphology is a result of opposing cycles of fusion and fission (Pernas & Scorrano, 2016). *MIC26* KO and DKO cells show a comparable extent of mitochondrial fragmentation, whereas *MIC27* KO show normal mitochondrial morphology indicating that perhaps mitochondrial defect of *MIC26* KO cannot be compensated by MIC27 and rather MIC26 acts independent of MIC27 in regulating mitochondrial morphology. Neither OPA1 levels nor the abundance of different forms were altered, indicating that mitochondrial fusion is normal in these cells. Further experiments are required to understand why and how MIC26 specifically regulates mitochondrial dynamics. On the other hand, we found that only DKO cells show reduced respiration, whereas *MIC26* KO cells show no change, and *MIC27* KO cells even show slight increase in respiration that could be attributed to compensation (or even enhancement) of the respiration defect by each other in SKOs of *MIC26* and *MIC27*.

The levels of respiratory chain complexes (RCs) or supercomplexes (SCs) were drastically reduced in DKO cells lacking *MIC26* and *MIC27*. The major respiratory chain complexes I, III, and IV are organized into supramolecular assemblies called supercomplexes (SCs) (Schägger & Pfeiffer, 2000). Although individual, isolated complexes are functional, formation of SCs is proposed to promote the stability of single complexes that enhances electron flow among them while reducing formation of ROS (Lapuente-Brun et al, 2013; Milenkovic et al, 2017). Experiments using mutants of tBid and acute ablation of OPA1 show that the cristae shape determines the stability and assembly of SCs (Cogliati et al, 2013). Here, we found that loss of CJs in *MIC26* and *MIC27* DKO cells determine the stability of the RCs or SCs. However, in this scenario, levels of OPA1 remain unaltered suggesting that these changes are independent of OPA1 regulation. CL is thought to act as a glue holding the respiratory chain (super) complex with several components of RCs which harbour CL-binding sites. Mutations in tafazzin (TAZ) that is required of remodelling of CL has been associated with Barth syndrome (Brady et al, 2006) and other mitochondrial deficiencies, whereas decreased amounts of cardiolipin are found in many human diseases, including diabetic cardiomyopathy. Deletion of *cardiolipin synthase* in *Drosophila* flight muscle causes aberrant cristae (Acehan et al, 2011). Barth syndrome patients' cells have altered cristae structure accompanied by destabilized RCs (McKenzie et al, 2006; Acehan et al, 2007), providing a molecular link between cristae structure and RCs formation. We found that overexpression of cardiolipin synthase (CRLS1) restored the stability of RCs and SCs and respiration in DKO cells showing that the reduced levels of cardiolipin in DKO cells directly affect the stability and integrity of RCs or SCs. However, the question remains how loss of CJs in conjunction with cristae defects affects the levels of CL or vice versa. Either the MICOS subunits (MIC26 and MIC27) directly influence the biosynthesis of cardiolipin or indirectly affect cardiolipin levels due to change in

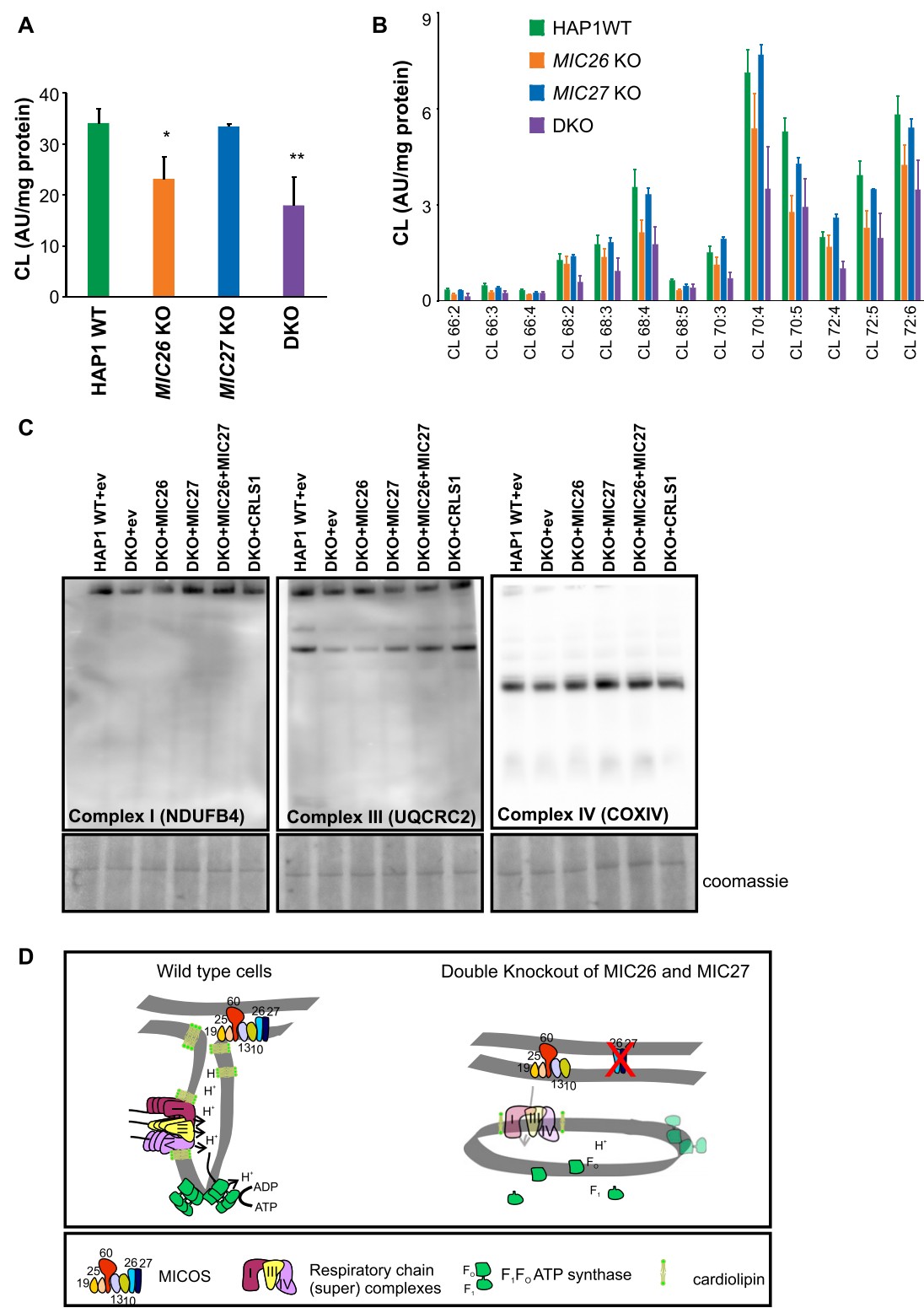

**Figure 7. MIC26 and MIC27 maintain cardiolipin levels that are required for stability of respiratory chain (super) complexes.**
**(A)** Graph representing the levels of cardiolipin shown as arbitrary units normalized to mg of protein in each cell types show significant reduction in MIC26 KO and double knockout (DKO) cells. *P-value ≤ 0.05, **P-value ≤ 0.01. t test was used for statistical analysis. **(B)** Graph showing the distribution of various cardiolipin species (arbitrary units normalized to mg of protein) in HAP1 WT, *MIC26* KO, *MIC27* KO, and DKO cells. **(C)** Blue-native gel electrophoresis for isolated mitochondria from HAP1 WT expressing empty vector (ev) and DKO cell lines that are stably expressing ev or MIC26 or MIC27 or MIC26 and MIC27 together or CRLS1 (cardiolipin synthase) were solubilized and blotted for antibodies specific for complex I (NDUFB4), complex III (UQCRC2), or complex IV (COX1V). The restoration of staining of respiratory chain (super)

cristae/CJs organization which might disrupt the membranes or supply of precursors required for cardiolipin synthesis. A large cardiolipin-synthesizing scaffold is present in mitochondria that interacts with MIC60 or MIC19 (Serricchio et al, 2018), and MICOS influences phospholipid synthesis (Harner et al, 2014; Aaltonen et al, 2016). In addition, intramitochondrial phospholipid transport in conjugation with MICOS is required for the formation of tubular cristae (Kojima et al, 2019). In our study, we propose that MIC26 and MIC27 being lipid-binding proteins of the MICOS provide the crucial interface between phospholipids such as CL and other scaffolding proteins to mediate formation of CJs and confer stability to RCs and SCs (Fig 7D).

A complex interplay between several cristae-shaping protein complexes is thought to sculpt the intricate cristae structures (Kondadi et al, 2019). While MICOS is required for the formation of highly curved CJs, rows of dimers of $F_1F_O$–ATP synthase complex are required for the formation of positive curvature at the rim of a crista. In yeast, Mic60/Fcj1 functions antagonistic to $F_1F_O$–ATP synthase for the formation of CJs and crista rims, respectively (Rabl et al, 2009). Mic10 binds to dimeric $F_1F_O$–ATP synthase, whereas Mic27 also binds to Su e (the yeast homolog of ATP5I), both promoting the oligomerization of $F_1F_O$–ATP synthase (Eydt et al, 2017; Rampelt et al, 2017a). In mammalian cells, OPA1 functionally interacts with $F_1F_O$–ATP synthase and favours its oligomerization (Patten et al, 2014; Quintana-Cabrera et al, 2018b). This prompted us to check the status of $F_1F_O$–ATP synthase oligomerization in our DKO cells lacking MIC26 and MIC27. There was no obvious defect in oligomerization of the $F_1F_O$–ATP synthase, but the overall amount of the whole complex (monomers and oligomers) was reduced, demonstrating a reduced stability or integrity of the $F_1F_O$–ATP synthase complex upon simultaneous deletion of MIC26 and MIC27. Our complexome data show a specific partial dissociation of subunits belonging specifically to the $F_1$ part of the complex from the monomeric complex in DKO cells, perhaps causing the instability and loss of the monomeric ($F_1F_O$) complex. This is different to baker's yeast where Mic27 rather influences the assembly and stability of the dimeric/oligomeric $F_1F_O$–ATP synthase complex (Eydt et al, 2017; Rampelt et al, 2017a), indicating an evolutionary divergence in regulation of the assembly of the $F_1F_O$–ATP synthase. Despite common evolutionary routes, the auxiliary factors required for assembly and regulation of $F_1F_O$–ATP synthase are not conserved among different organisms (Ruhle & Leister, 2015). INAC complex (comprising Ina22 and Ina17) helps in promoting the linkage between $F_1$ and $F_O$ in yeast, whereas its mammalian homolog is not yet identified (Lytovchenko et al, 2014; Naumenko et al, 2017).

The regulation of these apolipoproteins is highly complex because of the presence of a 55-kDa glycosylated form of MIC26 (MIC26$_{55kDa}$), which is usually secreted and found in blood plasma. MIC26$_{55kD}$ is elevated in the heart transcriptome of an animal model of diabetes (Lamant et al, 2006). Human patients of ACS have increased levels of MIC26$_{55kDa}$ in the plasma that correlated with an

independent inflammatory marker for ACS (Yu et al, 2012). Although MIC26$_{55kD}$ is present in high density lipoproteins, its in vivo function is not clear as it does not apparently influence any of the high density lipoproteins function tested. Although overexpression of Mic26 in a mouse heart caused ROS production and cardiac lipotoxicity (Turkieh et al, 2014), it leads to aggravated liver steatosis and accumulation of triglycerides in liver (Tian et al, 2017). It is possible that the effects observed upon the overexpression of Mic26 in mouse model or cardiomyopathic conditions could occur as a consequence of impaired mitochondrial respiratory machinery as we observed earlier (Koob et al, 2015) and here.

In summary, we find that apolipoproteins of the MICOS complex, MIC26 and MIC27, act cooperatively to regulate the formation of CJs and manage the stability and assembly of the RCs/SCs and $F_1F_O$–ATP synthase perhaps by modulating the cardiolipin levels. With a comprehensive functional analysis of cells after simultaneous deletion of MIC26 and MIC27 including a complexome approach, we revealed a novel cooperative function of these two proteins in determining the stability and integrity of the landscape of OXPHOS complexes. Further experiments are needed to provide mechanistic insights about how MIC26 and MIC27 affect OXPHOS complex biogenesis and cardiolipin levels and how this is linked to the pathophysiological role of these proteins in human diseases such as diabetic cardiomyopathy and mitochondrial myopathy (Beninca et al, 2020).

# Materials and Methods

### Cell culture

HAP1 WT, MIC26 KO, MIC27 KO, and DKO cells were obtained and custom-made by Horizon (UK) using the CRISPR-Cas method. The following guided RNA sequences were selected—for MIC26, TGAGG-GTCAATCGAAGTATG in exon 3 and for MIC27, ACAACCAGTTGCAG-TGCGGA in exon 3. MIC26 KO cells contain a 1-bp insertion in exon 3, whereas MIC27 KO cells contain an 8-bp deletion in exon 3. Later, MIC26 KO cells were used to target MIC27 with the same guided RNA that this time yielded a 160-bp deletion in the exon 3 of MIC27. The HAP1 cells were cultured using IMDM media supplemented with 20% fetal bovine serum and 1% penicillin and streptomycin. The cells were grown in an incubator at 37°C supplemented with 5% $CO_2$.

### Generation of stable cell lines using retroviral transduction

MIC26, MIC27, and CRLS1 were cloned into pMSCVpuro (PT3303-5; Clontech) using GIBSON cloning (E2611L; New England Biolabs). CRLS1 ORF was obtained from Sino Biological (HG20234-U). ORFs of MIC26 and MIC27 were taken from pcDNA3.1-Myc-MIC26 (Koob et al,

complexes compared with DKO (with ev) was found upon expression of MIC26 and MIC27 as well as CRLS1 in DKO cell lines. The part of the BN–PAGE stained with Coomassie is shown to represent the loading among the cell lines. **(D)** The scheme summarizing the phenotype occurring due to loss of MIC26 and MIC27 that show MIC26 and MIC27 are cooperatively required for the formation of crista junctions, maintenance of cardiolipin levels, and stability of respiratory chain (super) complexes and $F_1F_O$–ATP synthase. In addition, MIC26 and MIC27 are required for the assembly of $F_1F_O$–ATP synthase by facilitating the association of $F_1$ and $F_O$ part. Loss of MIC26 and MIC27 leads to impaired respiration.

2015) and pcDNA3.1-MIC27-FLAG (Weber et al, 2013). For retroviral transduction, Plat-E cells (kindly provided by Toshio Kitamura, Institute of Medical Science, University of Tokyo, Japan [Morita et al, 2000]) were plated in a 6-cm dish overnight and transfected with the respective plasmids (pMSCVpuro (ev), pMSCVpuro-MIC26, pMSCVpuro-MIC27, both pMSCVpuro-MIC26 and pMSCVpuro-MIC27 together, and pMSCVpuro-CRLS1) using FuGENE transfection reagent (Promega). After 48 h, the recombinant vesicular stomatitis virus-G pseudo-typed retroviruses were recovered from the supernatant of Plat-E cells after centrifugation and transferred to the target cells (HAP1 WT and DKO of *MIC26* and *MIC27*). After 72 h, the target cells were subjected to selection in puromycin (2 μg/ml) containing media to select the cells that stably express the transgene that confer puromycin resistance. The cell lines were confirmed by Western blots (MIC26 and MIC27) or sequencing (CRLS1).

### Electron microscopy

HAP1 WT, *MIC26* KO, *MIC27* KO, and DKO cells were grown on a petri dish and processed for electron microscopy as described earlier (Anand et al, 2016). Briefly, the cells were fixed using 3% glutaraldehyde in 0.1M sodium cacodylate buffer, pH 7.2, and subsequently pelleted. The cell pellets were embedded in agarose and stained with 1% osmium tetroxide for 50 min and 1% uranyl acetate/1% phosphotungstic acid for 1 h. The ultrathin sections were prepared using microtome, and imaging was performed on transmission electron microscope (H600; Hitachi) at 75 V equipped with Bioscan model 792 camera (Gatan) and analyzed with ImageJ software.

### SDS electrophoresis and Western blotting

For preparing the samples of Western blotting, the cells were collected in a small tube and proteins were extracted using RIPA lysis buffer. The amount of the solubilized proteins in each sample was estimated using the Lowry method (Bio-Rad). 15% SDS electrophoresis gel was used for running the protein samples. The proteins were subsequently blotted onto nitrocellulose membrane and probed with antibodies listed here, MIC13 (custom-made by Pineda; against human MIC13 peptide CKAREYSKEGWEYVKARTK), MIC27 (HPA000612; Atlas Antibodies), MIC26 (MA5-15493; Thermo Fisher Scientific), MIC60 (custom-made, Pineda; against human MIC60 using the peptide CTDHPEIGEGKPTPALSEEAS), MIC10 (ab84969; Abcam), MIC25 (20639-1-AP; Proteintech), β-tubulin (Cell Signaling Technology), and MIC19 (25625-1-AP; Proteintech). ATP5A (ab14748; Abcam), ATP5G (Abcam), ATP5D (ab97491; Abcam), ATP5IF1 (ab110277; Abcam), ATP5I (16483-1-AP; Proteintech), NDUFB4 (ab110243; Abcam), SDHB (ab14714; Abcam), UQCRC2 (ab14745; Abcam), COXIV (ab16056; Abcam), and OPA1 (custom-made, Pineda against human OPA1 using peptide CDLKKVREIQEKLDAFIEALHQEK, [Barrera et al, 2016]). The chemiluminescent signals were captured using VILBER LOURMAT Fusion SL (Peqlab). LI-COR Image studio software was used for quantification and image analysis.

### Respiration measurements

All the respiration measurements were performed using Seahorse XFe96 Analyzer (Agilent). The HAP1 cells were seeded in Seahorse XF96 cell culture plate (Agilent) at a density of $3 \times 10^4$ to $3.3 \times 10^4$ cells per well overnight. On the subsequent day, cells were washed and incubated in basic DMEM media (D5030; Sigma-Aldrich) supplemented with glucose, glutamine, and pyruvate at 37°C in non-$CO_2$ incubator 1 h before the assay. Mitochondrial respiration function was measured using Seahorse XF Cell Mito Stress Test kit (Agilent) according to the manufacturer's instructions. Briefly, the delivery chambers of the sensor cartridge were loaded with Oligomycin ($F_1F_o$–ATPase synthase inhibitor) or FCCP (uncoupler) or Rotenone and Antimycin (Complex I and Complex III inhibitor, respectively) to measure basal, proton leak, maximal, and residual respiration in XFe96 Analyzer. Cell number was normalized after the run using Hoechst staining. Data were analyzed using wave software (Agilent).

### Mitochondrial morphology imaging and quantification

HAP1 cells expressing matrix-targeted GFP were used to study the mitochondrial morphology on a PerkinElmer spinning disc confocal microscope equipped with a 60× oil objective (NA = 1.49) and a chamber maintaining 37°C and 5% $CO_2$. Images were acquired with a Hamamatsu C9100 camera having dimensions of 1,000 × 1,000 pixels after excitation at 488 nm. The cells were classified as tubular, intermediate, and fragmented depending on the majority of mitochondria present in a cell belonging to a particular class. Cells classified as intermediate class contained a mixture of predominantly short pieces, few tubular or fragmented mitochondria, whereas cells classified as tubular and fragmented contained mostly long tubular and very short fragments of mitochondria, respectively.

### Immunofluorescence staining

HAP1 and HeLa cells were fixed with 3.7% pre-warmed (37°C) paraformaldehyde for 20 min, washed thrice with PBS, permeabilized with 0.15% Triton X-100 and blocked using 10% goat serum for 15 min each. After blocking, the cells were incubated with respective primary antibodies overnight at 4°C and washed thrice. Secondary antibody incubation was performed at room temperature for 1 h followed by three washes with PBS. The samples were then used for STED imaging.

### STED super-resolution nanoscopy

Images were acquired using a 100× oil (NA = 1.4) or 93× glycerol (NA = 1.3) objective to acquire a field of 9.7 × 9.7 μm area (12× zoom for 100× and 12.9× zoom for 93× objectives, respectively) on Leica SP8 microscope fitted with a STED module. For HAP1 cells, primary antibodies were used against MIC10 (84969; Abcam), MIC60 (custom-made; Pineda), and MIC27 (HPA000612; Atlas Antibodies) after which goat antirabbit Abberior STAR 635P (Abberior) was used as the secondary antibody. Using a hybrid detector (HyD), images of HAP1 cells immunostained for MIC10, MIC60, and MIC27 were acquired at an emission range from 640 to 735 nm, whereas the signal was depleted using a pulsed STED depletion laser active at 775 nm. To increase the specificity of the signal, gating STED was active from 1 ns onward, where the images acquired had a pixel size of 17 nm. HeLa cells expressing MIC26-GFP were used as samples for staining

MIC26. Mouse Anti-GFP antibody (11814460001; Merck) and goat antimouse Alexa Fluor 488 (Thermo Fisher Scientific) were used as primary and secondary antibodies, respectively. Images were acquired from 495 to 585 emission range, whereas the signal was depleted using a continuous wave STED depletion laser active at 592 nm. Gating STED was used from 1.5 ns onward, whereas the images acquired had a pixel size of 21 nm. Images were processed as described earlier (Kondadi et al, 2020).

### Complexome analysis

Sample preparation, mass spectrometry, data analysis, and raw mass spectrometry data of complexome analysis have been deposited to the ProteomeXchange Consortium via the PRIDE partner repository (Perez-Riverol et al, 2019) with the dataset identifier PXD016733 and PXD016732. Averaged subunit quantification values were used for complex reference profiles. Reference profiles were normalized to maximum appearance between samples.

### Mitochondrial isolation and BN gel electrophoresis

For mitochondrial isolation, cells were pelleted at 600$g$ for 5 min and resuspended in an isotonic buffer containing 220 mM mannitol, 70 mM sucrose, 20 nM Hepes (pH 7.5, KOH), 1 mM EDTA and 1× protease inhibitor cocktail (Roche). The cells were mechanically ruptured by repeatedly passing through a syringe needle of 26 G cannula for 20 times. Debris or nuclei were separated by centrifugation at 1,000$g$ for 5 min. The supernatant was then centrifuged at 8,000$g$ for 10 min to collect the mitochondrial pellet. The protein estimation was performed using the Lowry method (Bio-Rad). Mitochondria were solubilized using 2 g/g of digitonin to protein ratio and equal amounts of mitochondria from each cell lines were loaded on a gradient gel (3–18%) and proceeded according to the method described earlier (Anand et al, 2016).

### In-gel activity of F$_1$F$_o$–ATP synthase

Isolated mitochondria were solubilized with increasing concentration of digitonin to protein ratio (in g/g) and were loaded on a 4–13% gradient gel to separate the macromolecular complexes. The gel slice was incubated in ATP synthase activity buffer (35 mM Tris, 270 mM glycine, 14 mM MgSO$_4$, 0.2% wt/vol Pb(NO$_3$)$_2$, and 8 mM ATP pH 8.3) at 25°C for 4 h and fixed using 50% methanol and transferred to water for imaging.

### qRT-PCR

For RNA isolation, 1 × 10$^6$ cells from each cell line were collected and homogenized using QIAshredder (QIAGEN), and RNA was extracted using RNeasy kit (QIAGEN) according to the manufacturer protocol. 1 $\mu$g of RNA was converted into cDNA using QuantiNova Reverse Transcription kit (QIAGEN). 15 ng of cDNA from each cell lines (HAP1 WT, *MIC26* KO, and *MIC27* KO) was used to perform qRT-PCR reaction using Quanti-Nova SYBR Green PCR kit (QIAGEN) with the following primers: MIC26 (forward: 5′-CCGTGAAGGTTGATGAGCTT reverse: 5′-GGAGC-TGTGAGATGCTTTCTT), MIC27 (forward: 5′-ATGCAGCCAAACAAGAGGAA reverse: 5′-GGAGCGGTGGTGCAGTAT), GAPDH (forward: 5′-CCCCGGTTTC-TATAAATTGAGC reverse: 5′-CGAACAGGAGGAGCAGAGAG), and HPRT1

(forward: 5′-CCTGGCGTCGTGATTAGTG reverse: 5′-TGAGGAATAAA-CACCCTTTCCA) in qRT-PCR Rotorgene 6000 (Corbett Research/QIAGEN). The analysis was performed using the Rotor-Gene Q 2.3.4 software for calculating 2$_T^{-\Delta\Delta C}$ (Livak & Schmittgen, 2001).

### Lipid analysis

Cell pellets containing 2 × 10$^6$ cells were extracted according to Matyash et al (2008). In brief, samples were homogenized using two beads (stainless steel, 6 mm) on a Mixer Mill (GER; 2 × 10 s, frequency 30/s; Retsch) in 700 $\mu$l methyl-tert-butyl ether/methanol (3/1, vol/vol) containing 500 pmol butylated hydroxytoluene, 1% acetic acid, and 150 pmol internal standard (IS; 18:3/18:3/18:3 triacylglycerol; Larodan). Total lipid extraction was performed under constant shaking for 30 min at room temperature. After addition of 140 $\mu$l dH$_2$O and further incubation for 30 min at room temperature, the samples were centrifuged at 1,000$g$ for 15 min. 500 $\mu$l of the upper, organic phase was collected and dried under a stream of nitrogen. Lipids were resolved in 500 $\mu$l 2-propanol/methanol/dH$_2$O (7/2.5/1, vol/vol/v) for UHPLC–Q-TOF analysis. The extracted cell proteins were dried and solubilized in 0.3 N NaOH at 65°C for 4 h and the protein content was determined using Pierce BCA reagent (Thermo Fisher Scientific) and BSA as standard.

Chromatographic separation was performed on a 1290 Infinity II LC system (Agilent) equipped with a Zorbax Extend-C18 rapid resolution HT column (2.1 × 50 mm, 1.8 $\mu$m; Agilent) running a 16-min linear gradient from 60% solvent A (H$_2$O; 10 mM ammonium acetate, 0.1% formic acid, 8 $\mu$M phosphoric acid) to 100% solvent B (2-propanol; 10 mM ammonium acetate, 0.1% formic acid, 8 $\mu$M phosphoric acid). The column compartment was kept on 50°C. A 6560 Ion Mobility Q-TOF mass spectrometer (Agilent) equipped with Dual AJS ESI source was used for detection of lipids in positive Q-TOF mode. Data acquisition was done by MassHunter Data Acquisition software (B.09; Agilent). Lipids were manually identified, and lipid data were processed using MassHunter Quantitative Analysis (B.09; Agilent). Data were normalized for recovery, extraction-, and ionization efficacy by calculating analyte/IS ratios (AU) and expressed as AU/mg protein.

### Statistics

$t$ test was used for comparison. In case of comparison between the values that were normalized to 1, one sample $t$ test was used. GraphPad prism software was used for statistical analysis and preparation of figures.

## Data Availability

The complexome profiling data from this publication has been deposited to ProteomeXchange Consortium via the PRIDE partner repository (Perez-Riverol et al, 2019) with the dataset identifier PXD016733 and PXD016732. The dataset is publicly available.

## Supplementary Information

# Acknowledgements

We thank Andrea Borchardt for excellent technical assistance in electron microscopy experiments and help in cell culture work. Experiments with STED were performed at the Centre for Advanced Imaging facility, Düsseldorf. This research is supported by research funding from the Research Committee of the Medical faculty of Heinrich Heine University Düsseldorf Foko-02/2015 (R Anand and AS Reichert) and FoKo-37/2015 (AK Kondadi) and Deutsche Forschungsgemeinschaft (DFG) grant RE 1575/2-1 (AS Reichert). Data were partly compiled within the Collaborative Research Center 1208: Identity and dynamics of membrane systems–from molecules to cellular functions funded by the DFG (German Research Foundation) – Project-ID 267205415 – Collaborative Research Center 1208 – project B12 (AS Reichert). I Wittig is supported by the DFG: Collaborative Research Center 815/Z1, EXC2026: Cardio Pulmonary Institute and by the Bundesministerium für Bildung und Forschung (BMBF) mitoNET—German Network for Mitochondrial Disorders 01GM1906D.

## Author Contributions

R Anand: conceptualization, data curation, formal analysis, supervision, investigation, visualization, methodology, project administration, and writing—original draft, review, and editing.

AK Kondadi: data curation, formal analysis, supervision, validation, investigation, visualization, and methodology.

J Meisterknecht: investigation and methodology.

M Golombek: investigation and methodology.

O Nortmann: investigation and methodology.

J Riedel: investigation and methodology.

L Peifer-Weiß: investigation and methodology.

N Brocke-Ahmadinejad: formal analysis, validation, investigation, visualization, and methodology.

D Schlütermann: investigation and methodology.

B Stork: supervision, investigation, and methodology.

TO Eichmann: resources, data curation, formal analysis, validation, investigation, visualization, and methodology.

I Wittig: resources, data curation, formal analysis, supervision, validation, investigation, visualization, and methodology.

AS Reichert: conceptualization, data curation, supervision, funding acquisition, investigation, project administration, and writing—original draft, review, and editing.

## Conflict of Interest Statement

The authors declare that they have no conflict of interest.

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
