## [Reviewer comments · Life Science Alliance]

Life Science Alliance

MIC26 and MIC27 cooperate to regulate cardiolipin levels and the landscape of OXPHOS complexes

Ruchika Anand, Arun Kumar Kondadi, Jana Meisterknecht, Mathias Golombek, Oliver Nortmann, Julia Riedel, Leon Peifer-Weiß, Nahal Brocke-Ahmadinejad, David Schlütermann, Björn Stork, Thomas O. Eichmann, Ilka Wittig, and Andreas Reichert

DOI: <https://doi.org/10.26508/lsa.202000711>

Corresponding author(s): Andreas Reichert, Heinrich Heine University Düsseldorf and Ruchika Anand, Heinrich Heine University Düsseldorf

Review Timeline:

Submission Date:	2020-03-20
Editorial Decision:	2020-03-23
Revision Received:	2020-07-11
Editorial Decision:	2020-07-16
Revision Received:	2020-07-20
Accepted:	2020-07-21

Transaction Report:

Please note that the manuscript was reviewed at *Review Commons* and these reports were taken into account in the decision-making process at Life Science alliance.

March 23, 2020

Re: Life Science Alliance manuscript #LSA-2020-00711

Prof. Andreas S. Reichert
Heinrich Heine University Düsseldorf
Institut für Biochemie und Molekularbiologie I
Universitätsstr. 1
Düsseldorf 40225
Germany

Dear Dr. Reichert,

Thank you for transferring your manuscript entitled "MIC26 and MIC27 act in synergy to regulate cardiolipin levels and the landscape of OXPHOS complexes" to Life Science Alliance. The manuscript was assessed by expert reviewers at Review Commons before, and you provided a point-by-point response to the concerns raised.

We appreciate your findings and based on the reviewer reports at hand and your revision plans, we would like to publish a revised version of your manuscript in Life Science Alliance. Please upload the final files using the link below, your login name is areichert1012

We are aware that many laboratories cannot function fully during the current COVID-19/SARS-CoV-2 pandemic and therefore encourage you to take the time necessary to revise the manuscript to the extent you outlined. We will extend our 'scoping protection policy' to the full revision period required. If you do see another paper with related content published elsewhere, nonetheless contact me immediately so that we can discuss the best way to proceed.

Thank you for this interesting contribution to Life Science Alliance. We are looking forward to receiving your revised manuscript.

Sincerely,

Andrea Leibfried, PhD
Executive Editor

Life Science Alliance
Meyerhofstr. 1
69117 Heidelberg, Germany
t +49 6221 8891 502
e a.leibfried@life-science-alliance.org
www.life-science-alliance.org

B. MANUSCRIPT ORGANIZATION AND FORMATTING:

Point-by-point reply to reviewers' comments

We thank all the reviewers for the critical comments that we believe has helped us further strengthen our manuscript. We provide the point-by-point response to your comments.

Reviewer #1 (Evidence, reproducibility and clarity (Required)):

The MICOS complex controls the architecture of the mitochondrial inner membrane, particularly the formation of cristae and cristae junctions. The complex contains two apolipoproteins MIC26 and MIC27. The function of these proteins is poorly understood. Anand and colleagues show in single knockout cell lines that both proteins exert partial overlapping functions. Whereas the single knockout cell lines display only minor effects, the double knockout of MIC26 and MIC27 in Hap1 cells reveals various defects, including onion-shape cristae, decreased cardiolipin levels and reduced stability of ATP synthase and respiratory chain complexes. Thus, MIC26 and MIC27 are both important to maintain full respiratory capacity and cardiolipin levels. The authors should address some minor points to strengthen their conclusions.

We thank the reviewer for overall positive feedback to our manuscript.

Comment 1: It remains unclear whether the reduced levels of cardiolipin affect the stability of respiratory chain complexes in the double knockout cell line or whether the loss of MIC26 and MIC27 more directly affect stability of these protein machineries. Does overexpression of the cardiolipin synthase leads to stabilization of the respiratory chain supercomplexes in the mutant cell line?

Response: We appreciate this excellent suggestion and have performed this experiment by stably overexpressing cardiolipin synthase (CRLS1) in double knockout cells of *MIC26* and *MIC27* using retroviral transduction method. CRLS1 was cloned in pMSCVpuro (retrovirus packaging vector) and was stably expressed in DKO cells that was confirmed by sequencing. As a control, *MIC26* and/or *MIC27* were also stably expressed in DKO cells shown in Figure S1A. We compared the staining of the respiratory chain complexes marked by antibodies specific for subunits of complex-I, complex-III or complex-IV in a blue native gel electrophoresis among DKO cells either overexpressing the empty vector (ev) or CRLS1 or *MIC26* and/or *MIC27*. We found that overexpression of CRLS1 in DKO cells restored the stability of respiratory chain complexes and supercomplexes (RCs and SCs), similar to simultaneous overexpression of *MIC26* and *MIC27*, indicating that reduced levels of cardiolipin in DKO cells directly affect the stability of RCs and SCs. This new data is included in the revised version as Figure 7C. Additionally, we also analysed whether overexpression of CRLS1 that restored stability of RCs and SCs could then rescue the respiration defect of DKO cells and indeed found a significant increase in respiration compared to DKO (+EV) cells after addition of CRLS1 in DKO (Figure S1B). These results indicate that reduced cardiolipin in DKO cells directly affects the stability of respiratory chain complexes as well as respiration in DKO.

Comment 2: The authors show that the cardiolipin levels are reduced in the absence of both MIC26 and MIC27. Are other phospholipid classes affected in the double knock out cell lines?

Response: This is a valid question and we would like to comment as follows. As cardiolipin is virtually exclusive to mitochondria, the changes that are observed in whole cell lysates can safely reflect the changes in the mitochondria. We used whole cell lysates (not purified mitochondria) to estimate the levels of cardiolipin in order to overcome the general problem of normalization of total protein/lipid content (of purified mitochondria) among different cell lines that have different membrane architecture and thus could potentially differ in mitochondrial purity due to differential centrifugation methods. The analysis of other phospholipid classes in our opinion only makes sense with purified mitochondria and appropriate normalization methods. This would require extensive fractionation and would encounter the issue of normalization among the cell lines. We believe that a global as well as an organelle-specific lipidomics approach is interesting but beyond the scope of the current manuscript.

Comment 3: As control, the authors should show whether re-expression of either MIC26 or MIC27 reverts the defects observed in the double knockout cell line.

Response: We agree that this is an important control experiment and therefore we generated stable cell lines that overexpresses MIC26 and/or MIC27 in double knockout cell lines and checked whether it could revert the defects of DKO. MIC26 or MIC27 separately or together were overexpressed in DKO cells as shown in Figure S1A. We analysed whether overexpression of MIC26 and MIC27 could rescue the reduced stability of respiratory chain supercomplexes and impaired respiration of DKO cells using blue native gel electrophoresis and oxygen consumption measurements. We found simultaneous re-expression of MIC26 and MIC27 could rescue the stability of respiratory chain complexes almost to the level that is comparable to control conditions (HAP1 WT+ empty vector). These results are included as Figure 7C in the revised version. Similarly, we found significant increase in mitochondrial respiration upon overexpression of both MIC26 and MIC27 in DKO cells. The overexpression of MIC26 was lower compared to the control conditions in these stable cell lines (Figure S1A) and could account of lower level of rescue (Figure S1B).

Comment 4: The authors should add a molecular weight marker to the blue native gels. Figure 4C: What is meant with "BMH"?

Response: We have now included molecular weight markers to the blue native gels in Fig 4C. BHM is bovine heart mitochondria that were used to determine the molecular weight in the blue native gels.

Comment 5: Figures 4C and S1: The large MICOS complex appears destabilized in the double knockout cell line, while lower MICOS forms remain largely unaffected. Does the parallel loss of MIC26 and MIC27 affect the integrity of MICOS?

Response: Thanks for pointing this out. We agree that there could be a slight decrease in large MICOS complex in DKO cells but this is a minor effect that could occur due to loss of MIC26 and MIC27 from the large MICOS complex as we clearly see that in WT cells MIC26 is abundantly present in this large MICOS complex. We now clearly mentioned this observation in result section of the revised manuscript and we further stated now that we cannot exclude that MIC26 and MIC27 may affect the integrity of the high molecular weight MICOS complex.

Reviewer #1 (Significance (Required)):

The MICOS complex is crucial to maintain cristae and cristae junctions. Analysis of protein machineries controlling membrane architecture is a rapidly developing field. The functions of MIC26 and MIC27 of the MICOS complex remain poorly understood. Here, the Reichert group provides a comprehensive characterization of single and double knockout cell lines of MIC26 and MIC27. Particularly, analysis of the double knockout cell line provides first insights into the role of these proteins for mitochondrial functions. The identified link between MIC26/MIC27 and cardiolipin levels and respiratory chain complexes is unexpected. The presented data are of high quality and all conclusions are well based on experimental results. The findings are very interesting for a broad readership and shed new light into the role of MIC26 and MIC27 for mitochondrial functions.

We thank the reviewer for highlighting the broader significance of our results on MIC26 and MIC27.

Reviewer #2 (Evidence, reproducibility and clarity (Required)):

The paper entitled "MIC26 and MIC27 act in synergy to regulate cardiolipin levels and the landscape of OXPHOS complexes" by Anand et al. sets out to study the relative contribution of MIC26 and MIC27 - two apolipoproteins that are members of the MICOS complex - to mitochondrial homeostasis.

To do so, the authors use MIC26 and MIC27 single and double knockout HAP1 cell lines. Using an array of assays, the authors conclude that MIC26 and MIC27 act synergistically, whereby loss of both proteins leads to disrupted mitochondrial function: including aberrant cristae, reduced ATPase assembly and lower cardiolipin levels.

Despite the breadth of approaches, there are several issues with the conclusions drawn from the data.

We thank the reviewer for the comments and provide the point-by-point response to explain the concerns raised.

****Major points****

Comment 1: A major concern with the data shown is that it appears that only one clone of the single and double knockout cells has been used. It is thus difficult to assess whether the phenotypes are caused by the intended genetic modification or by any other event that could happen during the isolation of the mutant clones. This, along with the lack of any rescue experiments, leads to some doubt in the robustness of the findings.

Response: We agree this is an important concern and is also shared by the other reviewers (comment 3 of reviewer 1 and comment 1 of reviewer 3,) and therefore we generated stable cell lines that overexpresses MIC26 and/or MIC27 in double knockout cell lines and checked whether it could revert defects occurring in DKO cells (see also the response to comment 3 of reviewer 1 for details). We analysed whether overexpression of MIC26 and MIC27 could rescue the reduced stability of respiratory chain supercomplexes and impaired respiration of DKO cells using blue native gel electrophoresis and oxygen consumption measurements. We found simultaneous re-expression of MIC26 and MIC27 could rescue the stability of respiratory chain complexes almost to the level that is comparable to control conditions (HAP1 WT+ empty vector). These results are included as Figure 7C in the revised version. Similarly, we found significant increase in mitochondrial respiration upon overexpression of both MIC26 and MIC27 in DKO cells. The overexpression of MIC26 was lower compared to the control conditions in these stable cell lines (Figure S1A) and could account of lower level of rescue (Figure S1B).

Comment 2: The authors conclude that loss of Mic26/27 causes a specific loss of higher order respiratory complexes. Interestingly, this loss in higher order assemblies is not accompanied by a concomitant increase in lower order assemblies (despite the total levels of the protein constituents of the complexes being unchanged upon loss of MIC26/27). It is not clear where the constituents of the complexes are now localised.

Response: We understand this concern, yet it can be explained by the technical limitation of blue-native gel electrophoresis, which is mainly used and optimized to reveal higher molecular weight protein complexes. This technique is not suited to detect unassembled proteins that are typically present at lower molecular weight. Moreover, we analysed a subset of subunits of each respiratory chain complexes in western blot analysis whose steady state levels were not altered in the KO cells. This strongly suggests that, at least the tested subunits, are not grossly degraded and are likely to be present as non-assembled subunits. Furthermore, we certainly cannot rule out the possibility that some key assembly factors are specially affected in knockout lines that cause the destabilization of respiratory chain complexes or super complexes. We have now clearly mentioned this possibility in the results section of the revised manuscript.

Comment 3: It is unclear to what extent the compensatory upregulation of either MIC26 or MIC27 in the single knockouts is rescuing any phenotypes. Indeed, the authors show in previous work (Koob et al., 2015) that acute loss of MIC26 has many effects on mitochondrial homeostasis. It could therefore be that MIC26 and MIC27 do not act synergistically, as proposed, but rather behave similarly and the increased protein levels of the other protein can compensate.

Response: Consistent from our earlier manuscript (Koob, Barrera et al., 2015), here we also find that MIC26 and MIC27 are reciprocally regulated such that overexpression or downregulation of one of them is always accompanied by the decrease or increase level of second protein respectively. Therefore, it is difficult to assign the individual function of one of them or determine the extent of compensation using the single depletion cell lines of MIC26 and MIC27. In this manuscript when we analysed the SKOs and DKO of *MIC26* and *MIC27*, we found that SKOs show moderate defects in cristae structure when compared to DKO that show severe cristae defects with increased appearance of onion-like cristae structure. Since SKOs still show cristae defects that are obviously only partially compensated with the respective other protein, we suggest that MIC26 and MIC27 act cooperatively because they are functionally overlapping, but not fully redundant. In other words, the lack of one subunit cannot be fully compensated by the other but it appears to be compensated to some extent. This partial complementation makes any analysis quite difficult, emphasizing the importance of the double knockout cell line analysed in detail in this study. We now modified the text in the discussion section to make this point clearer.

Comment 4: In the title, the authors said MIC26 and MIC27 act in synergy to regulate CL levels. But, there is no synergetic effect of the loss of Mic26 and 27 on cardiolipin level. Synergy is maybe an unfortunate choice of word.

Response: We understand this concern and therefore changed the word 'synergy' to 'cooperation' in the title and throughout the manuscript.

****Minor points****

Minor comment 1: The data showing that loss of MIC26 and MIC27 in the DKO cells has little effect on MICOS assembly (Fig 4a) would benefit from a positive control, i.e. a mutant in which MICOS is indeed disassembled. It is difficult to know what Mic60 or Mic 10 staining would look like in a situation where complex assembly is perturbed. It would probably still stain the IMM, maybe it would be less dotted? But without this knowledge, the interpretation is not straightforward. i.

Response: We agree to this point. We have performed such experiment in our recent publication where we found that deletion of MIC10 (that causes disassembly of MICOS and virtually complete loss of crista junctions) does not perturb the staining pattern of MIC60 (Kondadi, Anand et al., 2020). From this result, we concluded that MIC60 act as a pioneer for formation of crista junctions. In line with this, in this current study we now show that loss of MIC26 and MIC27 does not affect the staining of MIC10 and MIC60, reinforcing the claim that MIC10 and MIC60 are assembled earlier in MICOS than MIC26 and MIC27, that support the other results from this manuscript. We now included this new reference in the revised manuscript and adapted the text accordingly in the discussion section of the revised manuscript.

Minor comment 2: The Opa1 data in Figure S5 are not very convincing. It is concluded that there are no "obvious" changes to Opa1 but there are clear differences in levels between the genotypes and no quantification. It is unclear what an "obvious" change would be.

Response: We now included the quantification of OPA1 blots that show percentage of OPA1 in long or short forms do not change significantly among the cell lines. This is included in the revised version as Figure S6B.

Minor comment 3: Line 276-277, figure 5C I s referred to as Figure 5B in some parts of the text (page 10).

Response: Thank you for pointing this out. We changed this in the text.

Minor comment 4: Statistical analysis should be carried out not only between WT and other genetic backgrounds but also between SKO and DKO cells.

Response: We performed these additional statistical analyses and included the results in the respective figure legends of Figure 1. In most cases the differences among the SKOs and between SKO and DKO were non-significant except in between comparison of CJs number between MIC27 KO and DKO. These are included in the figure legends of the Figure 1.

Minor comment 5: Line 165-167, the quantitated levels of MIC26 or MIC27 should be provided (or refer to Fig 3C) to make the statement.

Response: Thanks for pointing this out. We included the reference to Fig 3C.

Minor comment 6: Fig 1C, it would be better to show the normal or abnormal mitochondrial morphologies in percentage. Mitochondrial section is unclear.

Response: Thanks for this excellent suggestion. We included this new quantification as Figure 1D. We omit 'mitochondrial section' and just used word 'per mitochondria' to avoid the confusion.

Minor comment 7: Fig. S4, the protein levels should be quantified. The blot herein only marginally supports the claim that "their levels were not significantly reduced in the DKO cells"

Response: We performed the quantification of protein levels in Figure S5 and included it in Figure S5B. No significant difference was found among HAP1 WT and DKO cells for all the antibody tested.

Minor comment 8: In Figure 7, the authors propose a model explaining that the simultaneous loss of MIC26 and MIC27 causes a reduction in cardiolipin levels, and this cause a destabilization of supercomplexes. However, it is currently unclear from the data if there is a causative link between these phenomena. There are arrows going from "loss of CJs" to "reduced cardiolipin" but no arrow going from "loss of CJs" to "reduced stability of RCs". Right now there is no data supporting this hierarchy.

Response: Thank you for pointing this. We did not intend to propose a strict hierarchy and therefore we now adapted the scheme to make it simpler.

Reviewer #2 (Significance (Required)):

The Micos complex is an important yet still mysterious protein complex on the inner mitochondrial membrane, involved in the maintenance of cristae junctions, contact sites with the Outer membrane, and maintenance of mitochondrial lipid homeostasis. The function of the different subunits is unclear. Two subunits Mic26 and 27 belong to the family of apolipoproteins, and knocking down one cause upregulation of the other, suggesting a compensatory mechanism. Here, Anand et al use CRISPR KO cells to inactivate mic26, 27 and both. They find that while these conditions do not appear to affect the assembly of the core MICOS complex, they had various effect on mitochondrial function, the double knock-out appearing to have the most severe effect. This indicate that Mic26 and 27 must be able somehow to partially compensate for each other's loss. As the MICOS complex appears

central to many aspects of inner-membrane biology, this manuscript represents an interesting advance for researchers in the field.

We thank the reviewer for conveying the interesting advance of our manuscript.

Reviewer #3 (Evidence, reproducibility and clarity (Required)):

Anand and co-workers investigate here the functional role of two subunits of the MICOS complex (MIC26 and MIC27) by generating human cell lines knock out for either each one individually of both simultaneously. Since the MICOS complex participate in the generation of the Cristae Junctions (CJs) at the mitochondrial inner membrane the study intend to rise light of the participation of these proteins in the CJ structure and function. The found that simultaneous deletion of both proteins induce aberrant cristae with reduce F1FO ATP synthase activity, reduced cell respiration, decreased steady-state levels of OXPHOS complexes and cardiolipin and partial dissociation of the F1FO ATP synthase. These set of observations led them to propose that MIC26 and MIC27 synergistically regulate cristae structure and the integrity of respiratory complexes and supercomplexes and the F1FO ATP synthase. They propose that they exert this action in an OPA1 independent way by modulating the levels of cardiolipin.

The work is nicely performed with high quality experimental analysis that support most of the conclusions and the paper is easy to read and clear. There are however some aspects that require clarification or complementary analysis to definitively sustain the conclusion derived from them.

We thank the reviewer for highlighting the high quality of our work. Here we provide the point-by-point explanation to each concern.

Comment 1: Generation of SKO and DKO cell lines. The information on methodological details of how the KO cell lines were generated is missing or this reviews fail to find it. No indication of the probes used, the evaluation of potential off-targets, etc. More detailed information on the generation of the KO cell is recommended. The DKO cell was generated from the MIC27KO. It would be desirable to generate a second DKO from the MIC26 to show the coincidence in the phenotype.

Response: Thank you for pointing out this. We now include the methodological details about the generation of these KO cell lines in the methods section. These cell lines were custom made by Horizon company upon request.

We understand the idea behind suggestion of generating another double knockout cell line from MIC26 KO to show the coincidence of the phenotype. To address this concern, we alternatively generated stable cell lines that overexpresses MIC26 and/or MIC27 in double knockout cell lines and checked whether it could revert the defects associated with DKO cells (see also the response to comment 3 of reviewer 1 for details). We analysed whether overexpression of MIC26 and MIC27 could recue the reduced stability of respiratory chain supercomplexes and impaired respiration of DKO cells using blue native gel electrophoresis and oxygen consumption measurements. We found simultaneous re-expression of MIC26 and MIC27 could recue the stability of respiratory chain complexes almost to the level that is comparable to control conditions (HAP1 WT+ empty vector). These results are included as Figure 7C in the revised version. Similarly, we found significant increase in mitochondrial respiration upon overexpression of both MIC26 and MIC27 in DKO cells. The overexpression of MIC26 was lower compared to the control conditions in these stable cell lines (Figure S1A) and could account of lower level of rescue (Figure S1B).

Comment 2: The TEM representative figures in Fig. 1C does not match with the quantitative data plotted in figure 1E-E. This is particularly true for the MIC27KO that barely shows differences with WT in terms of cristae shape and amount in the TEM micrograph but however this is not reflected in the quantitative analysis. It is recommended to double-check this discrepancy. In principle, this may only reflect the selection of not-the-best representative TEM image, however, MIC27KO cells showed activation of respiration (Fig 2A) and mitochondria elongation (Fig 2B) while MIC26KO does not seem to affect respiration (Fig 2A) despite the structural alteration in cristae and massive fragmentation. This should be commented. In addition, the discrepancy in functional and structural phenotypes between SKO raised concerns on the coincidence in the ultrastructural quantitative data showed in Fig 1D-E. Sorting this apparent discrepancy is important to define the relative role of MIC26 and MIC27 in the cristae structure. Data in Fig 2 may suggest that the depletion of MIC26 induce a strong phenotype but that of MIC27 does not, or even promote the opposite phenotype. Please clarify this point.

Response: We thank the reviewer for bringing out this discrepancy in selected images in Fig 1C. We accordingly changed the figures to 'more-representative' images for *MIC26* KO and *MIC27* KO. Additionally, now included another quantification classifying mitochondria in each cell lines to having 'abnormal' cristae (suggested by reviewer 2, minor comment 6) in Figure 1D. We believe that such quantification would help to bring about the changes that are observed in cristae structure in SKOs and DKO. We agree that *MIC26* KO does not show reduced respiration despite massive fragmentation and cristae defects and this is now clearly stated in this revised manuscript. In case of Figure 2, we would like to point out that we believe that *MIC26* acts independently to *MIC27* in regulating mitochondrial morphology as we do not see any severe increase in mitochondrial fragmentation in DKO compared to *MIC26* KO. We now highlight this point in the discussion.

Comment 3: The analysis of the stability of respiratory complexes and supercomplexes concluded that in the DKO cells both are significantly reduced. In comparing the analysis of the ATP synthase and the MICOS with that of the respiratory complexes the latter seemed rather superficial. The figures reflect differential changes induced by SKO with respect to DKO and also changes that affect in a different way RCs and SCs that are not commented by the authors. As an example Figure 5 and FS3 indicate that the distribution of RCs between free or SCs is clearly impacted by the deletion of *MIC27* (it seems to increase the relative amount of SCs). Interesting, is *MIC27* SKO the cell line showing an increase in respiration and hyper elongated mitochondria. In addition, the distribution of CII is also clearly modified in *MIC27* cells with additional high molecular weight positions that are clearly minority or inexistent in the other cell lines. The amount of data available should allow to comment on this issues.

Response: Thank you for bringing out this point. We have now expanded our description of these interesting aspects in the result section and adapted the text accordingly.

Comment 4: The conclusion of that OPA1 processing is not affected in the SKO or DKO cells is not supported by the figure S5. On the contrary, by eye is evident that the proportion of processed vs unprocessed OPA1 (S-OPA1 with respect L-OPA1) is increased in all mutant cells likely more in the DKO. These blots should be properly quantified. In view of the results shown the statement that the observed changes in SKO and DKO are independent of OPA1 is not supported.

Response: Thanks for this comment. This concern is also shared by reviewer 2 (minor comment 2), and we have performed the quantification of OPA1 blots in SKOs or DKO and data is shown in Fig S6B that shows no significant change in forms of OPA1 (long or short) among all the cell lines.

Reviewer #3 (Significance (Required)):

The manuscript present a interesting analysis of the role of MICOS in conforming the structure lot the mitochondrial cristae. It illustrate and confirm previous work showing that crista structure is determinant in the the organization of the the five OXPHOS complexes and its arrangement and super assembly. In this respect, the results will consolidate the relationship between shape and function of the mitochondrial inner membrane enriching the collection of protein components that participate on it. On top of that, the results define different levels of relevance of MICOS components to perform its role in shaping the mitochondrial inner membrane.

We thank the reviewer for placing the emphasise on the general significance of our work.

The field of expertise of this reviewer can be define us: OXPHOS, respiratory complexes and supercomplexes, mitochondria dynamics and function.

References:

Kondadi AK, Anand R, Hänsch S, Urbach J, Zobel T, Wolf DM, Segawa M, Liesa M, Shirihai OS, Weidtkamp-Peters S, Reichert AS (2020) Cristae undergo continuous cycles of membrane remodelling in a MICOS-dependent manner. *EMBO reports* 21: e49776
Koob S, Barrera M, Anand R, Reichert AS (2015) The non-glycosylated isoform of MIC26 is a constituent of the mammalian MICOS complex and promotes formation of crista junctions. *Biochimica et biophysica acta* 1853: 1551-63

July 16, 2020

RE: Life Science Alliance Manuscript #LSA-2020-00711-TR

Dr. Andreas S. Reichert
Heinrich Heine University Düsseldorf
Institut für Biochemie und Molekularbiologie I
Universitätsstr. 1
Düsseldorf 40225
Germany

Dear Dr. Reichert,

Thank you for submitting your revised manuscript entitled "MIC26 and MIC27 cooperate to regulate cardiolipin levels and the landscape of OXPHOS complexes". We would be happy to publish your paper in Life Science Alliance pending final revisions necessary to meet our formatting guidelines.

-please fill out the Electronic License Form

A. FINAL FILES:

B. MANUSCRIPT ORGANIZATION AND FORMATTING:

Sincerely,

Reilly Lorenz
Editorial Office Life Science Alliance
Meyerhofstr. 1
69117 Heidelberg, Germany
t +49 6221 8891 414
e contact@life-science-alliance.org
www.life-science-alliance.org

July 21, 2020

RE: Life Science Alliance Manuscript #LSA-2020-00711-TRR

Dr. Andreas S. Reichert
Heinrich Heine University Düsseldorf
Institut für Biochemie und Molekularbiologie I
Universitätsstr. 1
Düsseldorf 40225
Germany

Dear Dr. Reichert,

Thank you for submitting your Research Article entitled "MIC26 and MIC27 cooperate to regulate cardiolipin levels and the landscape of OXPHOS complexes". It is a pleasure to let you know that your manuscript is now accepted for publication in Life Science Alliance. Congratulations on this interesting work.

DISTRIBUTION OF MATERIALS:

Again, congratulations on a very nice paper. I hope you found the review process to be constructive and are pleased with how the manuscript was handled editorially. We look forward to future exciting submissions from your lab.

Sincerely,

Reilly Lorenz
Editorial Office Life Science Alliance
Meyerhofstr. 1
69117 Heidelberg, Germany
t +49 6221 8891 414
e contact@life-science-alliance.org
www.life-science-alliance.org